# Phosphoglycerate kinase 1 acts as a cargo adaptor to promote EGFR transport to the lysosome

Shao-Ling Chu [1], Jia-Rong Huang[1], Yu-Tzu Chang[1], Shu-Yun Yao[1], Jia-Shu Yang[2], Victor W. Hsu [2] ✉ & Jia-Wei Hsu [1,3] ✉

The epidermal growth factor receptor (EGFR) plays important roles in multiple cellular events, including growth, differentiation, and motility. A major mechanism of downregulating EGFR function involves its endocytic transport to the lysosome. Sorting of proteins into intracellular pathways involves cargo adaptors recognizing sorting signals on cargo proteins. A dileucine-based sorting signal has been identified previously for the sorting of endosomal EGFR to the lysosome, but a cargo adaptor that recognizes this signal remains unknown. Here, we find that phosphoglycerate kinase 1 (PGK1) is recruited to endosomal membrane upon its phosphorylation, where it binds to the dileucine sorting signal in EGFR to promote the lysosomal transport of this receptor. We also elucidate two mechanisms that act in concert to promote PGK1 recruitment to endosomal membrane, a lipid-based mechanism that involves phosphatidylinositol 4,5-bisphosphate [PI(4,5)P2] and a protein-based mechanism that involves hepatocyte growth factor receptor substrate (Hrs). These findings reveal an unexpected function for a metabolic enzyme and advance the mechanistic understanding of how EGFR is transported to the lysosome.

Transport to the lysosome represents a major way of regulating the function of proteins on the cell surface by promoting their degradation. This process is critical for a variety of cellular events, including growth, differentiation, motility, and signal transduction[1–4]. Upon endocytosis, surface proteins are transported to the early endosome, where they can be either recycled to the cell surface or transported to the late endosome. For the latter fate, cargo proteins are sorted into internal vesicles of the late endosome (also known as the Multivesicular Body, MVB), resulting in their targeting to the lysosome[5–7].

Sorting into intracellular pathways involves cargo adaptors recognizing specific motifs on cargo proteins known as sorting signals[8–11]. Sorting of endosomal cargoes for transport to the lysosome is best characterized for the role of the ESCRT (endosomal sorting complex required for transport) complex. Components of this complex recognize ubiquitin as the sorting signal on endocytic proteins for their sorting into the internal vesicles of the late endosome, which then targets proteins for lysosomal degradation. The ESCRT complex is comprised of sub-complexes, known as ESCRT-0, −1, −2, and −3. They act sequentially starting at the early endosome in coupling cargo binding and membrane deformation, which ultimately results in endosomal cargoes being sorted into the internal vesicles of the late endosome[12,13].

One of the best characterized endocytic proteins that undergo lysosomal transport is the epidermal growth factor receptor (EGFR)[5,7,14,15]. Components of the ESCRT complexes have been found to interact with ubiquitin-modified EGFR for its sorting to the lysosome[16–18]. However, targeting to the lysosome has also been found

[1]Institute of Biochemical Sciences, National Taiwan University, Taipei 10617, Taiwan. [2]Division of Rheumatology, Inflammation and Immunity, Brigham and Women's Hospital, and Department of Medicine, Harvard Medical School, Boston, MA 02115, USA. [3]Institute of Biological Chemistry, Academia Sinica, Taipei 11529, Taiwan. ✉e-mail: vhsu@bwh.harvard.edu; jwhsu@ntu.edu.tw

to require a dileucine-based sequence in EGFR[19,20], but a cargo adaptor that recognizes this sorting signal has not been identified.

Phosphoglycerate kinase 1 (PGK1) is a metabolic enzyme that acts in glycolysis. In addition to this traditional role, PGK1 has been discovered more recently to act as a protein kinase in tumorigenesis and autophagy[21,22]. A previous study found that PGK1 is phosphorylated at serine position 203 (S203) by extracellular signal-regulated kinase (ERK) upon EGFR activation, which promotes the recruitment of PGK1 to mitochondria membrane to inhibit mitochondrial pyruvate metabolism[22]. However, whether this phosphorylation also recruits PGK1 to other cellular membranes is unknown. Here, we find that the S203 phosphorylation recruits PGK1 to endosomal membranes where it acts as a cargo adaptor that recognizes the dileucine sorting signal in EGFR to promote its transport to the lysosome. We also show that this phosphorylation promotes PGK1 recruitment to endosomal membranes in two complementary ways, a protein-based mechanism that involves binding to hepatocyte growth factor receptor substrate (Hrs) and a lipid-based mechanism that involves binding to PI(4,5)P$_2$. These findings not only reveal a previously unknown role of a metabolic enzyme, but also advance the mechanistic understanding of how endocytic EGFR is transported to the lysosome.

## Results

### PGK1 promotes EGFR transport to the lysosome

We previously discovered that glyceraldehyde 3-phosphate dehydrogenase (GAPDH), a metabolic enzyme that acts in glycolysis, can act non-canonically by regulating intracellular transport pathways[23]. Thus, as PGK1 is another glycolytic enzyme that has been found to have non-canonical roles[21,22], we explored whether it could also act in intracellular transport.

We initially performed a screen of the major intracellular pathways using a quantitative microscopy-based approach that tracks model cargoes for specific pathways, as previously described[23]. Briefly, for the early secretory system, we examined anterograde transport from the endoplasmic reticulum (ER) to the Golgi by tracking an ER pool of Vesicular Stomatitis Virus G protein (VSVG), and retrograde transport from Golgi to the ER by tracking a Golgi pool of VSVG-KDELR (KDEL receptor). For the late secretory system, we examined anterograde transport from the Golgi to the plasma membrane (PM) by tracking a Golgi pool of VSVG, and retrograde transport from the PM to the Golgi by tracking the fate of cholera toxin B subunit (CTxB) after binding at the cell surface. We found that small interfering RNA (siRNA) against PGK1 had no appreciable effect on these pathways [Fig. 1a–d and Supplementary Fig. 1a–d].

For the endocytic pathways, we examined endocytosis by tracking the internalization of surface EGFR to the early endosome (Fig. 1e and Supplementary Fig. 1e), and endocytic recycling by tracking the transport of the transferrin receptor (TfR) from the recycling endosome to the PM (Fig. 1f and Supplementary Fig. 1f). We found that siRNA against PGK1 also had no appreciable effect on these pathways (Fig. 1e, f and Supplementary Fig. 1e, f). However, examining endosomal transport to the lysosome, by tracking the fate of EGFR that had been internalized from the cell surface, we observed delay in this transport induced by the siRNA treatment (Fig. 1g and Supplementary Fig. 1g). Consistent with this finding, PGK1 overexpression had the opposite effect of accelerating EGFR transport to the lysosome (Fig. 1h and Supplementary Fig. 1h).

CXCR4 is another model cargo that undergoes endocytic transport to the lysosome[24]. We found that siRNA against PGK1 does not inhibit this transport (Fig. 1i and Supplementary Fig. 1i). TfR also undergoes endocytic transport to the lysosome upon treatment with ferric ammonium citrate (FAC)[25]. Assessing this transport, we found that siRNA against PGK1 does not have an appreciable effect (Supplementary Fig. 1j). Consistent with this finding, TfR degradation was also

not significantly affected (Fig. 1j). Thus, PGK1 shows relative specificity in regulating EGFR transport to the lysosome.

We also confirmed the efficacy of the siRNA treatment against PGK1 by immunoblotting for endogenous PGK1 (Fig. 1j). Furthermore, as HeLa cells were used for the transport screen, we examined A549 cells and found that siRNA against PGK1 inhibits EGFR transport similarly (Supplementary Fig. 2a). Thus, the role of PGK1 in EGFR transport is not cell-specific. We also ruled out that the inhibition of EGFR transport induced by siRNA against PGK1 could be attributed to effects on lysosomes, as this treatment did not affect the number of lysosomes (Supplementary Fig. 2b).

As inhibition of transport to the lysosome should reduce the degradation of endocytic proteins, we next confirmed that siRNA against PGK1 reduced the degradation of endocytic EGFR (Fig. 2a, b). Consistent with this finding, the total level of EGFR was increased in siRNA-treated cells (Supplementary Fig. 2c). Moreover, EGFR signaling, as tracked by ERK activation, was prolonged (Fig. 2a, c).

We also confirmed the specificity of the siRNA targeting, as two different siRNA sequences against PGK1 had similar effects in reducing EGFR degradation and prolonging ERK activation (Fig. 2a). Moreover, the overexpression of PGK1 resulted in the opposite effect of enhancing EGFR degradation (Fig. 2d, e) and reducing ERK activation (Fig. 2d, f). Furthermore, consistent with our finding above that siRNA against PGK1 does not affect CXCR4 transport to the lysosome, we found that CXCR4 degradation was not affected by this siRNA treatment (Fig. 2g).

We next treated cells with siRNA against PGK1 followed by rescue with either the wild-type or the catalytic-dead (T378P) form of PGK1 to achieve physiologic levels of both proteins (Supplementary Fig. 2d). We found that the expression of the catalytic-dead PGK1 (T378P) resulted in EGFR being transported to the lysosome similarly as the wild-type form (Fig. 2h). Moreover, both conditions showed similar degrees of EGFR degradation (Supplementary Fig. 2e, f) and ERK activation (Supplementary Fig. 2e, g). Thus, the catalytic activity of PGK1 is not required for its ability to promote EGFR transport.

### PGK1 acts as a cargo adaptor to promote EGFR transport to the lysosome

To gain insight into how PGK1 promotes EGFR transport through a non-catalytic role, we next performed a co-precipitation experiment on cell lysates and found that EGF stimulation enhances the association of PGK1 with EGFR (Fig. 3a). We then performed a pulldown assay using purified components and found that PGK1 can interact directly with the cytoplasmic domain of EGFR, specifically the juxta-membrane region (Fig. 3b and Supplementary Fig. 3a). Moreover, the catalytic-dead mutant of PGK1 interacted with this region similarly as the wild-type form (Fig. 3c).

We next noted that a dileucine motif located within this region of EGFR has been found previously to act as a sorting signal to promote the transport of endosomal EGFR to the lysosome[19,20]. However, a cargo adaptor predicted to recognize this sorting signal remains unknown. We first confirmed that mutating the dileucine motif inhibited EGFR transport to the lysosome (Fig. 3d and Supplementary Fig. 3b), as well as reducing EGFR degradation (Fig. 3e, f) and enhancing ERK activation (Fig. 3e, g). We also confirmed that the mutation still enabled EGFR to be transported to the cell surface (Supplementary Fig. 3c). We then found that mutating this dileucine motif prevents PGK1 from binding directly to EGFR (Fig. 3h). As control, mutating a different dileucine sequence within this region did not affect this direct interaction (Fig. 3h).

We next performed a competition assay using purified components and confirmed that a peptide that contains the relevant dileucine motif prevents PGK1 from binding directly to EGFR, while a peptide having the dileucine motif mutated shows impaired competition (Fig. 3i). Furthermore, co-precipitation studies confirmed that PGK1 does not associate with EGFR in cells when the dileucine sorting signal

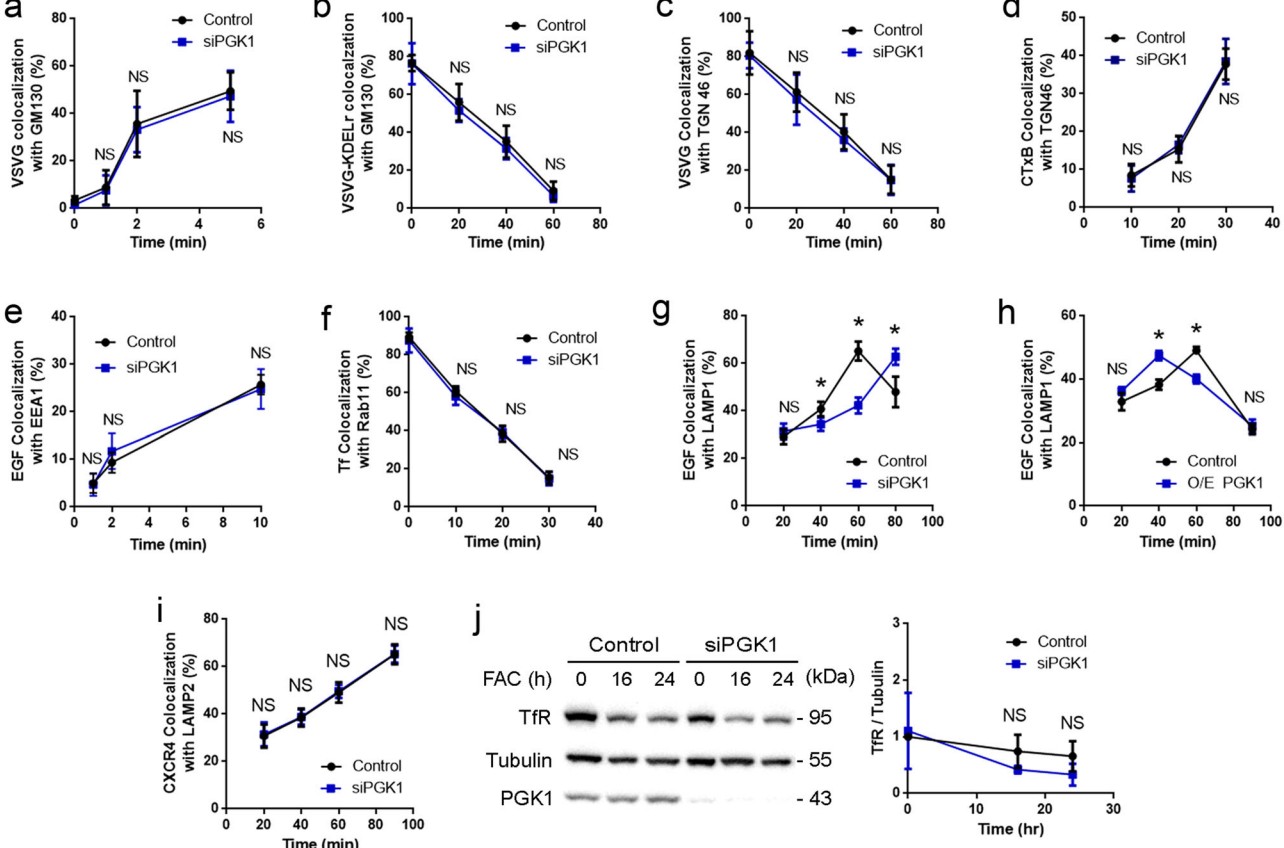

**Fig. 1 | PGK1 promotes EGFR transport to the lysosome.** Quantitative results are shown as mean ± s.e.m.; *$p < 0.05$, NS (not significant) $p > 0.05$, unpaired two-sided Student's $t$ test. **a** Colocalization of VSVG with a GM130 was performed, $n = 10$ cells examined over 3 independent experiments. Statistics is shown for the 0, 1, 2, and 5-min time point, $P = 0.485, 0.685, 0.694$, and $0.3059$, respectively. **b** Colocalization of VSVG-KDELR with GM130 was performed, $n = 10$ cells examined over 3 independent experiments. Statistics is shown for the 0, 20, 40, and 60-min time point, $P = 0.938, 0.238, 0.1303$, and $0.061$, respectively. **c** Colocalization of VSVG with TGN46 was performed, $n = 10$ cells examined over 3 independent experiments. Statistics is shown for the 0, 20, 40, and 60-minute time point, $P = 0.741, 0.546, 0.1244$, and $0.948$, respectively. **d** Colocalization of CTxB with TGN46 was performed, $n = 10$ cells examined over 3 independent experiments. Statistics is shown for the 10, 20, and 30-min time point, $P = 0.5119, 0.3485$, and $0.3766$, respectively. **e** Colocalization of EGF with EEA1 was performed, $n = 10$ cells examined over 3 independent experiments. Statistics is shown for the 1, 5, and 10-min time point,

$P = 0.808, 0.092$, and $0.2744$, respectively. **f** Colocalization of Tf with Rab11 was performed, $n = 10$ cells examined over 3 independent experiments. Statistics is shown for the 0, 10, 20, and 30-minute time point, $P = 0.4079, 0.1493, 0.5788$, and $0.3328$, respectively. **g** Colocalization of EGF with LAMP1 was performed, $n = 10$ cells examined over 3 independent experiments. Statistics is shown for the 20, 40, 60, and 80-min time point, $P = 0.1066, 8.4 \times 10^{-4}, 2.08 \times 10^{-11}$, and $6.9 \times 10^{-5}$, respectively. **h** Colocalization of EGF with LAMP1 was performed, $n = 10$ cells examined over 2 independent experiments. Statistics is shown for the 20, 40, 60, and 80-min time point, $P = 0.218, 0.0006, 0.00048$, and $0.654$, respectively. **i** Colocalization of CXCR4 with LAMP2 was performed, $n = 15$ cells examined over 3 independent experiments. Statistics is shown for the 20, 40, 60, and 90-min time point, $P = 0.2962, 0.4448, 0.6859$, and $0.9378$, respectively. **j** Total TfR level was quantified, $n = 4$. Statistics was performed for the 0, 16, and 24-h time point, $P = 0.8127, 0.203$, and $0.1873$, respectively.

is mutated (Fig. 3j). This result was also confirmed by the proximity ligation assay (PLA) that assesses protein interactions in intact cells (Supplementary Fig. 3d). Thus, the collective results suggested that PGK1 acts as a cargo adaptor by binding to the dileucine sorting signal in EGFR to promote its transport to the lysosome.

This newly defined role of PGK1 is likely to be independent of its traditional role in glycolysis, as the catalytic activity of PGK1 is not needed (see Fig. 2h). Further supporting this conclusion, we found that glucose starvation, which should reduce PGK1 function in glycolysis, still allowed EGF stimulation to induce PGK1 to associate with EGFR (Supplementary Fig. 3e). Moreover, altering the cellular level of PGK1, either by reducing its level through siRNA treatment or increasing its level through overexpression, had minimal effect on the cellular ATP level (Supplementary Fig. 3f), in contrast to the more appreciable effects that these perturbations had on EGFR transport. This difference could be explained by the cargo-binding function of PGK1, which is predicted to require stoichiometric protein level, while its glycolytic function would only need catalytic amount.

## PGK1 phosphorylation promotes its recruitment to endosomal membrane

As PGK1 needs to be recruited from the cytosol to endosomal membrane for its role as a cargo adaptor, we next investigated how this recruitment occurs. Led by a previous finding that the ERK-induced phosphorylation of PGK1 at its S203 residue promotes its recruitment to mitochondrial membrane[22], we tested whether PGK1 is also recruited to other cellular membranes. We fractionated cells into endosomal membrane and cytosol and found that PGK1 is recruited to endosomal membrane upon EGF stimulation, and this recruitment is inhibited by treatment with an ERK inhibitor (Fig. 4a). We also confirmed that this recruitment involves PGK1 being phosphorylated at the S203 residue, as assessed by an antibody that detects the phosphorylation of this residue in PGK1 (Fig. 4b).

To further confirm that this phosphorylation induces PGK1 recruitment to endosomal membrane, we next mutated the S203 residue to alanine (S203A), which prevents phosphorylation at this residue, or to aspartic acid (S203D), which mimics constitutive

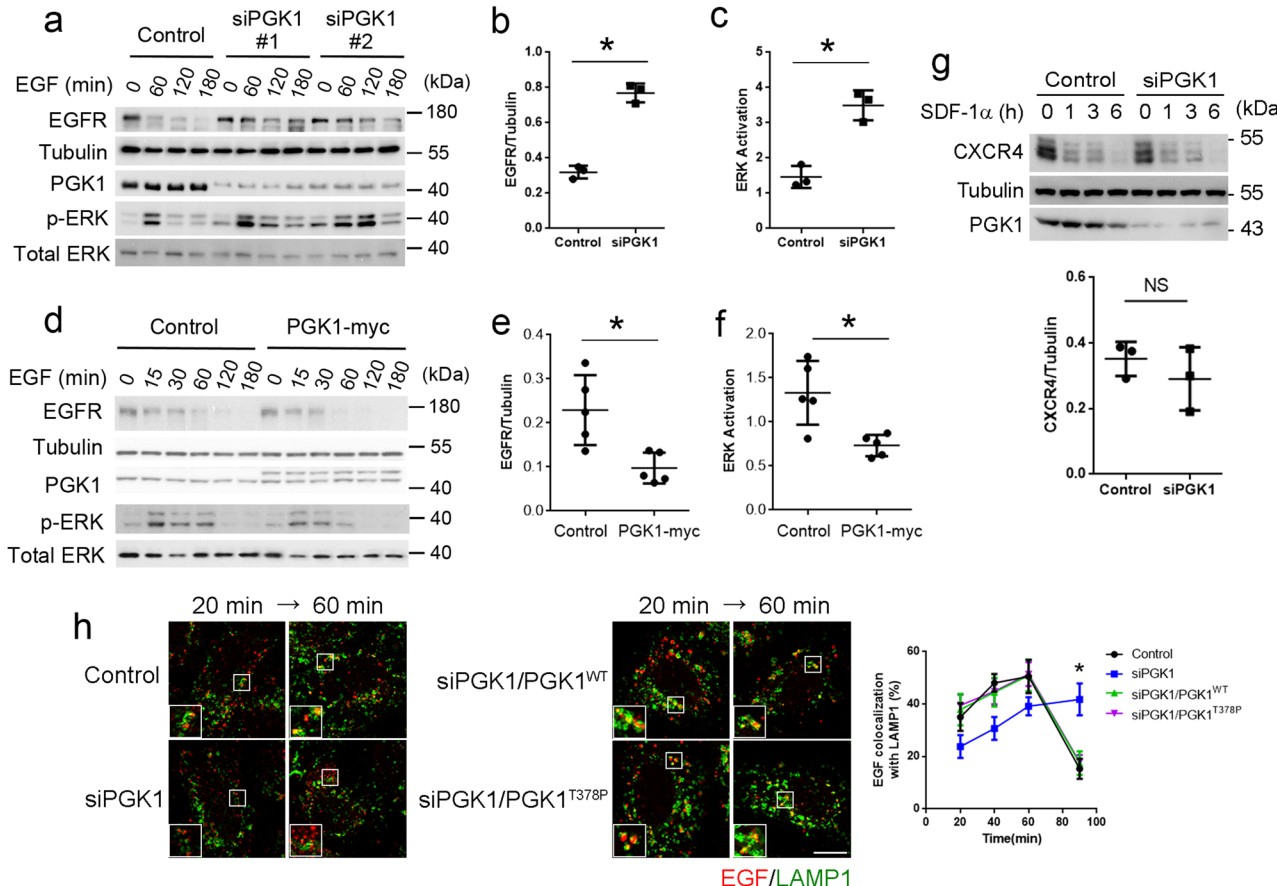

**Fig. 2 | PGK1 promotes EGFR degradation.** Quantitative results are shown as mean +/- s.e.m.; *$p < 0.05$, NS $p > 0.05$, unpaired two-sided Student's $t$ test. **a** Time-course analysis of EGFR degradation upon EGF treatment in HeLa cells. The effect of siRNA against PGK1 is examined. Cell lysates were immunoblotted for proteins indicated, $n = 3$. Two different siRNAs against PGK1 show similar results, with a representative experiment shown. Sequence #1 was used for all other experiments that involve siRNA against PGK1. **b** Quantitation of EGFR level at the 120-min time point for the analysis above that used siRNA #1 to reduced PGK1 level. EGFR level was normalized to tubulin level, $P = 4.27 \times 10^{-5}$. **c** Quantitation of ERK activation at the 120-min time point for the analysis above that used siRNA #1 to reduced PGK1 level. Phosphorylated ERK level was normalized to total ERK level, $P = 0.017$. **d** Time-course analysis of EGFR degradation upon EGF treatment for indicated time point in HeLa cells, examining the effect of PGK1 overexpression. Cell lysates were immunoblotted for proteins indicated, $n = 5$. A representative result is shown.

**e** Quantitation of EGFR level at the 60-min time point for the analysis above. EGFR level was normalized to tubulin level, $P = 0.003$. **f** Quantitation of ERK activation at the 60-min time point for the analysis above. Phosphorylated ERK level was normalized to total ERK level, $P = 0.0039$. **g** Time-course analysis of CXCR4 degradation upon SDF-1α treatment for indicated time point in HeLa cells, examining the effect of siRNA against PGK1. Cell lysates were immunoblotted for proteins indicated, $n = 3$. A representative result is shown above. Quantitation is shown below for the 3-h time point, $P = 0.1703$. CXCR4 level was normalized to tubulin level. **h** Assay for endocytic transport of EGFR to the lysosome, comparing the effect of expressing wild-type versus catalytic-dead form of PGK1. Colocalization of EGF with LAMP1 was performed, $n = 30$ cells examined over 3 independent experiments. Representative images are shown on left with EGF in red and LAMP1 in green, bar = 10 μm. Quantitation is shown on right with statistical analysis performed for the 90-min time point, $P = 0.9868$.

---

phosphorylation at this residue. We expressed physiologic levels of these phospho-mutants by treating cells with siRNA against PGK1 followed by rescue with the mutants (Fig. 4c). We found that the S203D mutant showed enhanced recruitment to endosomal membrane as compared to the S203A mutant (Fig. 4d).

We next found that the S203D mutant showed enhanced association with EGFR, while the S203A mutant exhibited the opposite behavior, as assessed by both a co-precipitation study (Fig. 4e) and PLA analysis (Fig. 4f). Consistent with these findings, we found that the expression of the S203D mutant enhanced EGFR transport to the lysosome, while the expression of the S203A mutant inhibited this transport (Fig. 4g). Moreover, the expression of the S203D mutant enhanced EGFR degradation, while the expression of the S203A mutant inhibited this degradation (Fig. 4h, i). Expression of the S203D mutant also reduced ERK activation while S203A expression promoted this activation (Fig. 4h, j).

We next examined whether the S203 phosphorylation enhances a direct interaction between PGK1 and EGFR. Performing a pulldown

experiment using purified components, we found that both phosphorylation mutants of PGK1 binds similarly to the juxta-membrane domain of EGFR (Fig. 4k). We also confirmed that mutating the dileucine sorting signal prevents this binding (Fig. 4k). Thus, as the S203 phosphorylation does not affect the direct interaction between PGK1 and EGFR, we next examined whether this phosphorylation promotes PGK1 recruitment to endosomal membrane.

We initially performed time-course analysis and found that the phosphorylation of the S203 residue of PGK1 can be detected after 15 min of EGF stimulation and peaks at 30 min (Supplementary Fig. 4a). PLA analysis revealed that the interaction between PGK1 and EGFR also peaks at 30 min (Supplementary Fig. 4b). We next pursued confocal microscopy and found that EGF stimulation results in EGFR localizing mainly at the early endosome at 15 min (Supplementary Fig. 4c, f), at the late endosome at 30 min (Supplementary Fig. 4d, h), and at the lysosome at 60 minutes (Supplementary Fig. 4e, j). Thus, these results suggested that PGK1 has maximal interaction with EGFR at the late endosome.

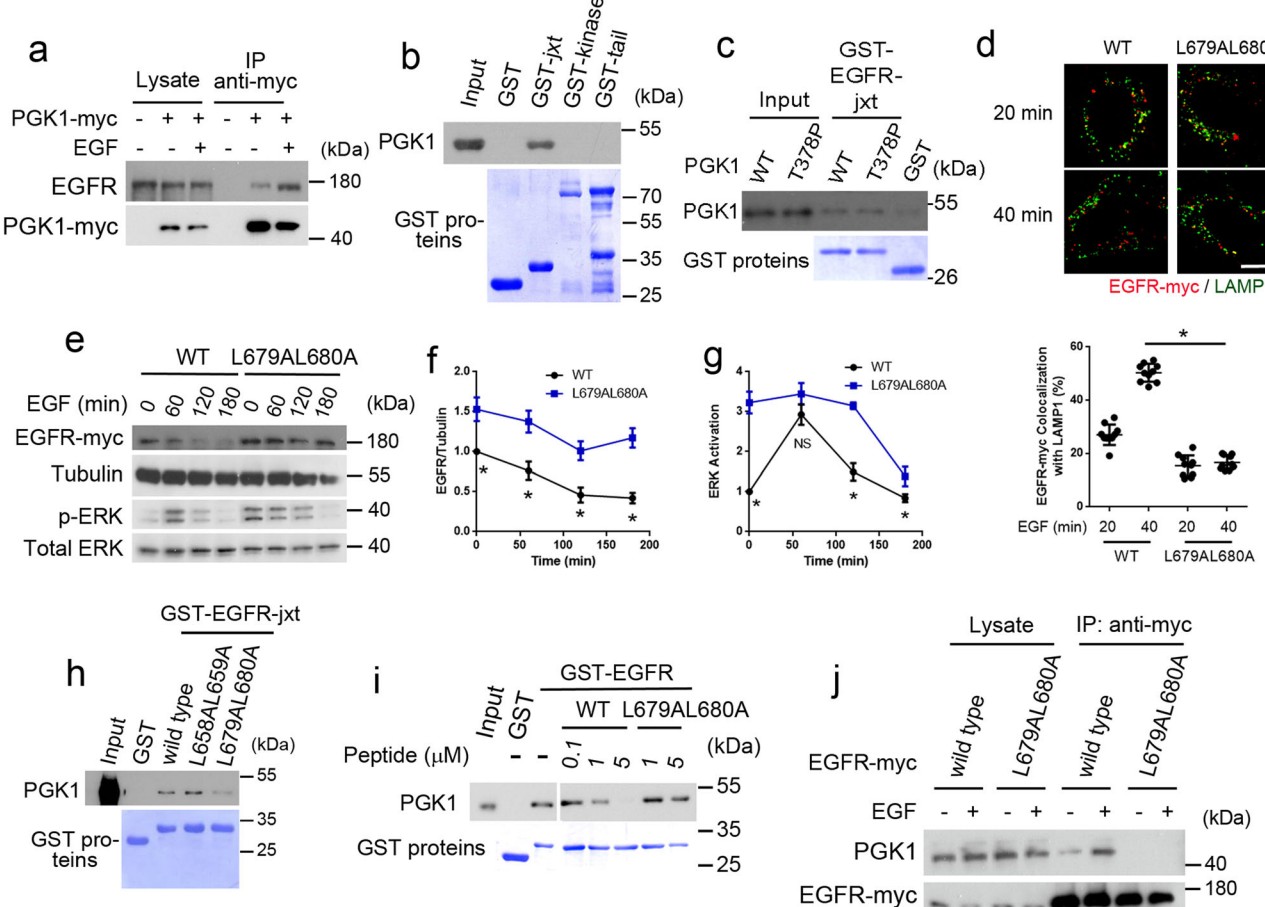

**Fig. 3 | PGK1 recognizes the dileucine sorting signal in EGFR.** Quantitative results are shown as mean +/- s.e.m.; *$p < 0.05$, unpaired two-sided Student's $t$-test. **a** Co-precipitation analysis examining the association of PGK1 with EGFR upon EGF stimulation for 1 h, $n = 3$. **b** Pull-down analysis examining the interaction of PGK1 to different forms of EGFR fused to GST, $n = 3$. **c** Pull-down analysis examining the interaction of different PGK1 forms with the juxta-membrane region EGFR fused to GST, $n = 3$. **d** Assay for endocytic transport of EGFR to the lysosome, examining the effect of mutating the dileucine sorting signal in EGFR. Colocalization of EGFR-myc with LAMP1 upon EGF treatment for indicated time point was performed, $n = 10$ cells examined over 3 independent experiments. Representative images are shown above with EGFR-myc in red and LAMP1 in green, bar = 10 μm. Quantitation is shown below for the 40-min time point, $P = 1.13 \times 10^{-15}$. **e** Time-course analysis of EGFR degradation upon EGF treatment for indicated time point in HeLa cells, examining the effect of mutating the dileucine sorting signal in EGFR. Cell lysates

were immunoblotted for proteins indicated, $n = 3$. **f** Quantitation of EGFR level at the 0, 60,120, and 180-min time point for the analysis above. EGFR level was normalized to tubulin level, $P = 0.0033, 0.004, 3.41 \times 10^{-4}$, and 0.0007, respectively. **g** Quantitation of ERK activation at the 0, 60,120, and 180-min time point for the analysis above. Phosphorylated ERK level was normalized to total ERK level, $P = 0.00014, 0.0741, 1.26 \times 10^{-4}$, and 0.023, respectively. **h** Pull-down analysis examining the interaction of PGK1 to various forms of EGFR juxta-membrane regions fused to GST as indicated. Purified components were used to examine direct interaction, $n = 3$. **i** Pull-down analysis examining the effect of titrating increased level of a peptide containing either the dileucine sorting signal of EGFR or this signal mutated on the direct interaction with PGK1, $n = 3$. **j** Co-precipitation analysis examining of the effect of mutating the dileucine sorting signal in EGFR on its association with PGK1. HeLa cells were stimulated with EGF for 1 h, $n = 3$.

We then noted a technical issue. At any given time-point, when the quantitative colocalizations of EGFR with endosomal markers were added together, the sum was greater than 100%. A reconciling explanation was suggested by a recent EM study that had found endosomal markers to have overlapping distributions[26]. Indeed, we found by confocal microscopy that significant colocalization existed between EEA1 and Rab7 (Supplementary Fig. 4l), and between Rab7 and Lamp1 (Supplementary Fig. 4m). Moreover, the use of confocal microscopy suggested a second contributing explanation, which was that the various endosomal compartments may not be completely distinguishable at the light level. Thus, when considering that the purpose of the time-course study was simply to assess the relative distribution of EGFR at different times after EGF stimulation, we next expressed the colocalization values instead as Pearson's coefficients (Supplementary Fig. 4g, i, k), in order to overcome the issue of marker overlap that results in the fraction of PGK1 colocalizing with endosomal markers being misleadingly high.

We next examined the localization of PGK1 upon EGF stimulation and found that it localized mainly at the early endosome at 15 min (Supplementary Fig. 5a), at the late endosome at 30 min (Supplementary Fig. 5b), and at the lysosome at 60 min (Supplementary Fig. 5c). Thus, these results suggested that the S203 phosphorylation targets PGK1 to endosomal compartments, starting at the early endosome and peaking at the late endosome. We also confirmed the specificity of staining for PGK1, as siRNA against PGK1 markedly reduced this staining (Supplementary Fig. 5d). Moreover, similar staining was observed using two different antibodies (Supplementary Fig. 5e), and GFP-tagged PGK1 showed similar distribution as staining by antibody (Supplementary Fig. 5f).

### Hrs mediates a protein-based mechanism of PGK1 recruitment to endosomal membrane

We next considered that the ESCRT complex has been well-established to act in the endosomal sorting of EGFR to the lysosome[5–7]. Thus, we

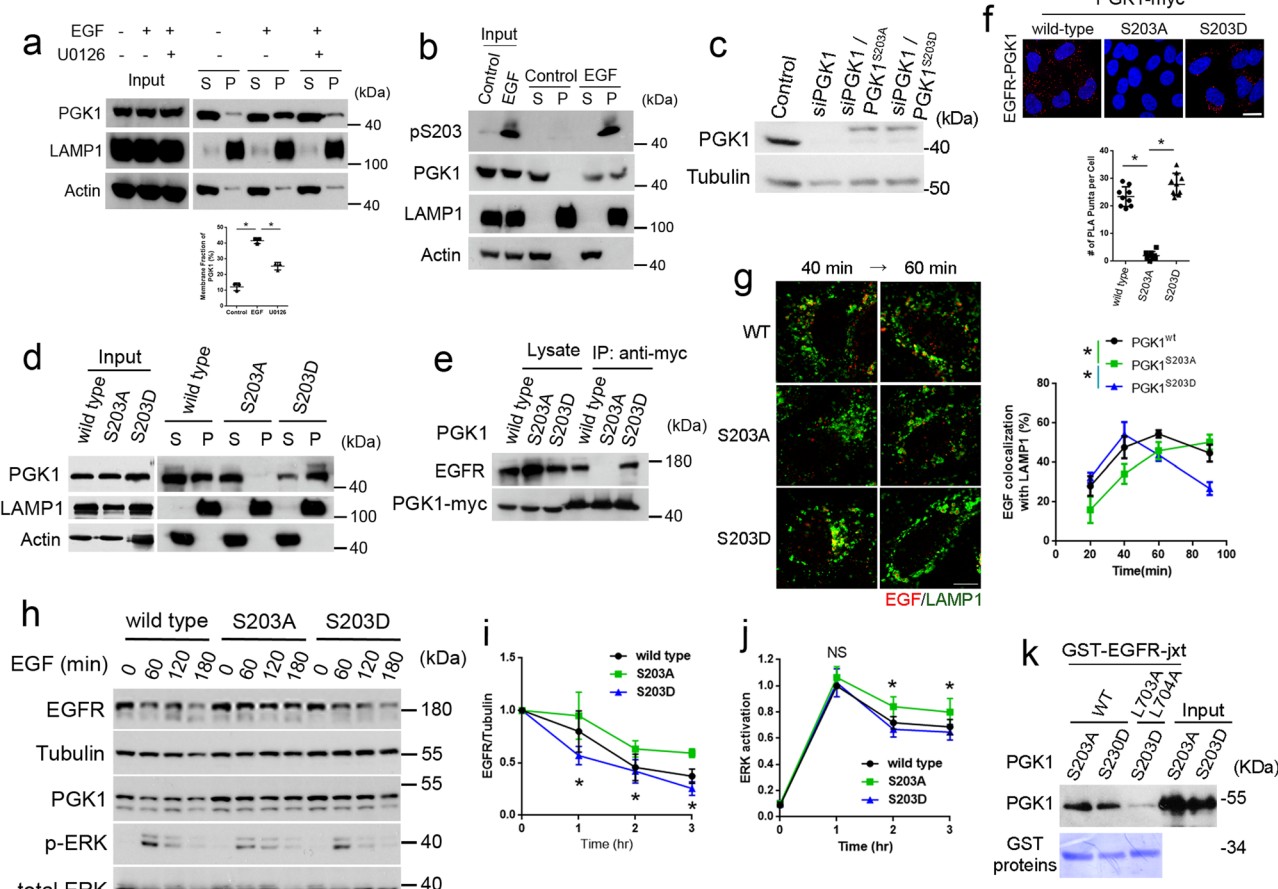

**Fig. 4 | S203 phosphorylation of PGK1 promotes EGFR transport to the lysosome.** Quantitative results are shown as mean +/- s.e.m.; *$p < 0.05$, unpaired two-sided Student's $t$ test. **a** Fractionation of cells into membranes containing endosomes (P) and cytosol (S) upon EGF stimulation for 30 min, with or without the ERK inhibitor U0126, $n = 3$. Quantitation is shown below, $P = 0.00014$ (control versus EGF) and 0.00041 (EGF versus U0126). **b** Fractionation of cells into membranes containing endosomes (P) and cytosol (S) upon EGF stimulation for 30 min, $n = 3$. **c** Immunoblotting of HeLa cell lysates examining the efficacy of siRNA against PGK1 and the level of rescue with phospho-mutant forms of PGK1, $n = 3$. **d** Fractionation of cells into membranes containing endosomes (P) and cytosol (S) upon EGF stimulation for 30 min, $n = 3$. **e** Co-precipitation analysis examining the association of wild-type and phospho-mutants of PGK1 with EGFR upon EGF stimulation for 30 min, $n = 3$. **f** PLA analysis examining the association of wild-type and phospho-mutants of PGK1 with EGFR upon EGF stimulation for 30 min, $n = 10$ cells examined over 3 independent experiments. Quantitation is shown for a representative experiment, $P = 1.1 \times 10^{-7}$ (wild-type versus S203A) $P = 2.6 \times 10^{-7}$ (S203A versus S203D). **g** Colocalization of EGF (red) with LAMP1 (green) upon EGF treatment for indicated time point was performed, $n = 20$ cells examined over 3 independent experiments, bar = 10 μm. Statistics is shown on right for the 40-min time point, $P = 8.265 \times 10^{-7}$ (wild-type versus S203A) $2.472 \times 10^{-11}$ (S203A versus S203D). **h** Time-course analysis of EGFR degradation upon EGF treatment for indicated time point, examining the effect of expressing PGK1 wild-type and phospho-mutants, $n = 3$. **i** Quantitation of EGFR level at the 60, 120, and 180-min time point for the analysis above. EGFR level was normalized to tubulin level, $P = 0.0192$, 0.0485, and 0.0352, respectively. **j** Quantitation of ERK activation at the 60, 120, and 180-min time point for the analysis above. Phosphorylated ERK level was normalized to total ERK level, $P = 0.5193$, 0.0025, and 0.0364, respectively. **k** Pull-down analysis examining the interaction of PGK1 phospho-mutants with the juxta-membrane region of EGFR (wild-type or dileucine mutated) fused to GST, $n = 3$.

sought to determine the relationship between the roles of ESCRT and PGK1 in EGFR transport. Focusing on ESCRT components known to interact with EGFR, which include Hrs, His domain-containing protein tyrosine phosphatase (HD-PTP) and Charged multivesicular body protein 4B (CHMP4B)[16,17], we found siRNA against PGK1 did not affect the association of EGFR with Hrs (Fig. 5a), HD-PTP (Fig. 5b), or CHMP4B (Fig. 5c), as assessed by co-precipitation analysis. However, siRNA against Hrs inhibited the association of PGK1 with EGFR (Fig. 5d), which was further confirmed by PLA analysis (Fig. 5e). PLA analysis also revealed that targeting against other components of the ESCRT complex did not affect the association of PGK1 with EGFR (Fig. 5e). Thus, these results identified Hrs to be required for PGK1 to associate with EGFR.

We next performed co-precipitation studies and found that EGF stimulation promotes the interaction between PGK1 and Hrs (Fig. 6a). Consistent with this finding, EGF stimulation enhanced the

colocalization between PGK1 and Hrs (Fig. 6b). Moreover, co-precipitation studies revealed that the S203D mutant shows enhanced interaction with Hrs, while the S203A mutant shows the opposite behavior (Fig. 6c). PLA analysis further confirmed these behaviors of the PGK1 phospho-mutants (Fig. 6d).

In light of the above findings, we next examined whether Hrs promotes PGK1 recruitment to endosomal membrane. We initially performed confocal microscopy and found that the S203D mutant showed enhanced colocalization with Hrs, while the S203A mutant showed reduced colocalization with Hrs (Fig. 6e). Moreover, siRNA against Hrs selectively reduced the localization of PGK1 at the late endosome (Supplementary Fig. 6b), as compared to that at the early endosome (Supplementary Fig. 6a) and at the lysosome (Supplementary Fig. 6c). We then examined PGK1 recruitment to endosomal membrane more directly. We isolated endosomal membrane depleted of Hrs, which was accomplished by treating cells with siRNA against

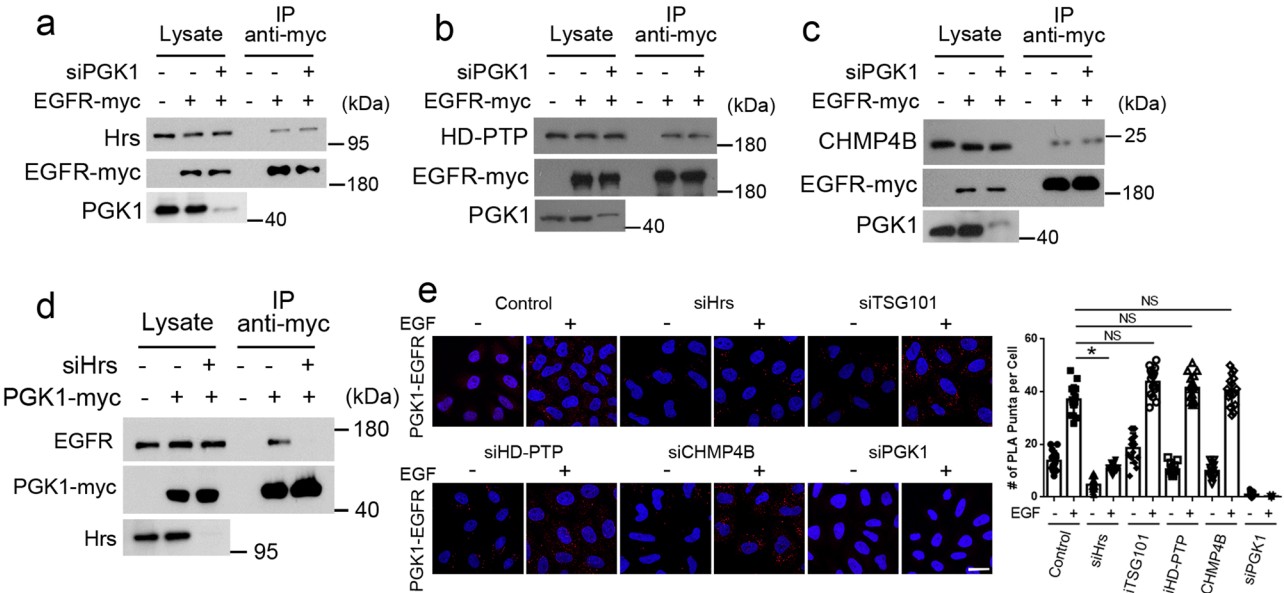

**Fig. 5 | Hrs is required for PGK1 to interact with EGFR.** Quantitative results are shown as mean +/- s.e.m.; *$p < 0.05$, NS $p > 0.05$, unpaired two-sided Student's *t* test. **a** Co-precipitation analysis examining the effect of siRNA against PGK1 on the association of Hrs with EGFR upon EGF stimulation for 15 min, $n = 3$. **b** Co-precipitation analysis examining the effect of siRNA against PGK1 on the association of HD-PTP with EGFR upon EGF stimulation for 15 min, $n = 3$. **c** Co-precipitation analysis examining the effect of siRNA against PGK1 on the association of CHMP4B with EGFR upon EGF stimulation for 15 min, $n = 3$. **d** Co-precipitation analysis examining the effect of siRNA against Hrs on the association of PGK1 with EGFR upon EGF stimulation for 30 min, $n = 3$. **e** PLA analysis examining the effect of targeting against different proteins through siRNA treatment on the association of PGK1 with EGFR in HeLa cells upon EGF stimulation for 30 min, $n = 15$ cells examined over 3 independent experiments. Puncta tracks the association of PGK1 with EGFR by using primary antibodies directed against endogenous PGK1 and EGFR, bar = 10 μm. Quantitation is shown on right for a representative experiment, $P = 5.93 \times 10^{-11}$ for si-Hrs, $P = 0.215$ for si-TSG101, $P = 0.0765$ for si-HD-PTP, and $P = 0.423$ for si-CHMP4B.

Hrs and then isolating endosomal membrane from these cells. We first confirmed that endosomal membrane was enriched by this procedure (Supplementary Fig. 6d). We then found that PGK1 recruitment to this membrane was reduced (Fig. 6f). Thus, these results suggested one mechanism by which the S203 phosphorylation promotes the endosomal recruitment of PGK1, by enhancing its interaction with Hrs.

We next performed EM analysis and found that siRNA against PGK1 reduced the level of EGFR in the internal vesicles of the late endosome (Supplementary Fig. 6e, with quantitation shown in left graph). Moreover, siRNA against PGK1 reduced the ratio of EGFR in internal vesicles versus those on the limiting membrane of the late endosome (Supplementary Fig. 6e, with quantitation shown in right graph). Thus, these findings suggested that PGK1 promotes EGFR transport not only from the early endosome to the late endosome, but also the sorting of EGFR into the internal vesicles of the late endosome.

### PIP5K1A mediates a lipid-based mechanism of PGK1 recruitment to endosomal membrane

We next noted that phosphatidylinositol 4,5-bisphosphate [PI(4,5)P$_2$] generated by phosphatidylinositol phosphate 5-kinase type Iγ (also known as PIP5K1C) has been found previously to promote EGFR transport to the lysosome[27]. Thus, we explored the possibility that, besides the protein-based mechanism of PGK1 recruitment that involves Hrs, a lipid-based mechanism that involves PI(4,5)P$_2$ could also be needed for efficient PGK1 recruitment. We initially found that siRNA against PIP5K1C did not appreciably affect the association of PGK1 with EGFR, but notably, siRNA against another PIP5K1 isoform, PIP5K1A, reduced this association (Fig. 7a). We then found that EGF stimulation promotes the association of PIP5K1A with PGK1 and EGFR, as assessed by co-precipitation studies (Fig. 7b).

To further characterize the role of PIP5K1A, we next performed confocal microscopy and found that EGF stimulation also induced the colocalization of PIP5K1A with EGFR at time points when the receptor

was transport through the endosomal compartments (Supplementary Fig. 7a). Furthermore, siRNA against PIP5K1A inhibited PGK1 from localizing to the early endosome (Supplementary Fig. 7b), late endosome (Supplementary Fig. 7c) and the lysosome (Supplementary Fig. 7d). This siRNA treatment also inhibited EGFR transport to the lysosome (Fig. 7c). Consistent with this delayed transport, siRNA against PIP5K1A inhibited EGFR degradation (Fig. 7d, e) and enhanced ERK activation (Fig. 7e, f). Moreover, the overexpression of PIP5K1A had the opposite effects, enhancing EGFR transport to the lysosome (Supplementary Fig. 8a) and EGFR degradation (Supplementary Fig. 8b, c), as well as reducing ERK activation (Supplementary Fig. 8b, d). We also found that the catalytic activity of PIP5K1A is needed for these effects of overexpression, as the overexpression of a catalytic-dead mutant (DNRQ) of PIP5K1A had the opposite effects of delaying EGFR transport to the lysosome (Supplementary Fig. 8e) and EGFR degradation (Supplementary Fig. 8f, g), as well as enhancing ERK activation (Supplementary Fig. 8f, h).

To confirm that PI(4,5)P$_2$ generation underlies how PIP5K1A promotes PGK1 recruitment to endosomal membrane, we next sought to reconstitute this recruitment. Incubating endosomal membrane with recombinant forms of PGK1, we initially confirmed that the S203D mutant was more efficiently recruited to this membrane than the wild-type form (Fig. 7g). We next depleted PIP5K1A from this membrane, which was achieved by treating cells with siRNA against PIP5K1A and then collected endosomal membrane from these cells. Using this membrane, we found that the enhanced PGK1 recruitment induced by the S203D mutation became impaired (Fig. 7h)

We next confirmed that PIP5K1A-depleted endosomal membrane had reduced level of PI(4,5)P2 (Supplementary Fig. 8i, left graph), and as control the level of PI(3,4)P2 was not affected (Supplementary Fig. 8i, right graph). Moreover, feeding cells with PI(4,5)P2, but not PI(3,4)P2, increased the level of PI(4,5)P2 in PIP5K1A-depleted endosomal membrane (Supplementary Fig. 8j, left graph), while feeding

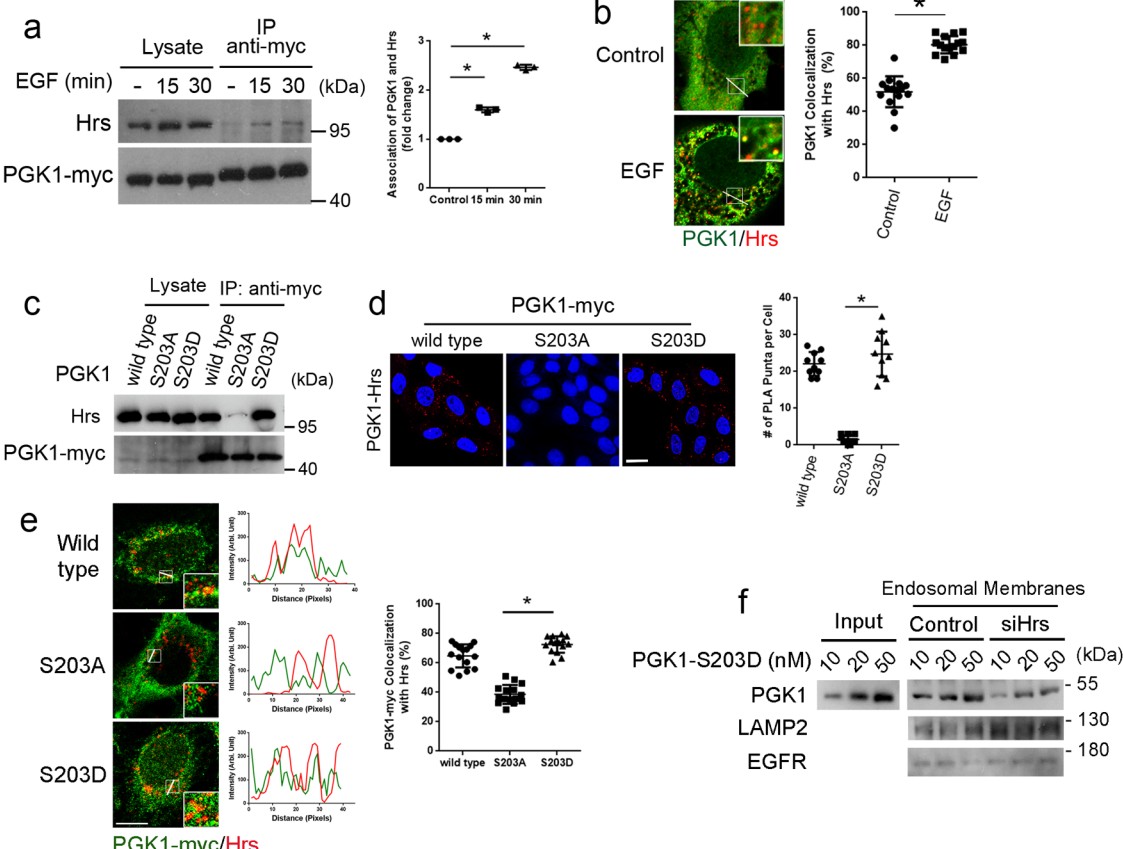

**Fig. 6 | Hrs promotes PGK1 recruitment to endosomal membrane.** Quantitative results are shown as mean +/- s.e.m.; *$p < 0.05$, unpaired two-sided Student's $t$ test. **a** Co-precipitation analysis examining PGK1 association with Hrs upon EGF stimulation at time points indicated, $n = 3$. Quantitation from three experiments is shown on right, $P = 0.00222$ for 15 min and $4.22 \times 10^{-4}$ for 30 min. **b** Colocalization of PGK1 (red) and Hrs (green) upon EGF stimulation for 30 min, $n = 15$ cells examined over 3 independent experiments, bar = 10 μm. Quantitation is shown on right for a representative experiment, comparing control versus EGF stimulation, $P = 2.387 \times 10^{-11}$. **c** Co-precipitation analysis examining PGK1 wild-type and phospho-mutants associating with Hrs upon EGF stimulation for 30 min, $n = 3$. **d** PLA analysis examining PGK1 wild-type and phospho-mutants associating with Hrs upon EGF stimulation for 30 min, $n = 10$ cells examined over 3 independent

experiments. Puncta tracks the association of PGK1 with Hrs by using primary antibodies directed against myc epitope of transfected PGK1-myc and endogenous Hrs, bar = 10 μm. Quantitation is shown on left for a representative experiment, $P = 5.7 \times 10^{-8}$. **e** Colocalization of PGK1 (green) wild-type and phospho-mutants with Hrs (red) upon EGF stimulation for 30 min, $n = 15$ cells examined over 2 independent experiments, bar = 10 μm. Quantitation is shown on right, $P = 2.816 \times 10^{-8}$ for S203A and S203D. **f** Reconstitution of PGK1 recruitment to endosomal membrane, examining the effect of depleting Hrs from the membrane. The S203D mutant of PGK1 was incubated with endosomal membrane that had Hrs depleted followed by immunoblotting for PGK1 on membrane. Immunoblotting for EGFR and LAMP2 assessed the level of membrane examined, $n = 3$.

---

cells with PI(3,4)P2, but not PI(4,5)P2, increased the level of PI(3,4)P2 in endosomal membrane (Supplementary Fig. 8j, right graph). We then found that PI(4,5)P$_2$, but not PI(3,4)P2, delivery to PIP5K1A-depleted membrane restored the recruitment of PGK1-S203D (Fig. 7i). Thus, the results altogether identified another mechanism by which the S203 phosphorylation promotes PGK1 recruitment, enhancing the ability of PGK1 to recognize PI(4,5)P$_2$ generated by PIP5K1A on endosomal membrane.

## Discussion

We have discovered that PGK1 promotes EGFR transport to the lysosome and have also elucidated how this unexpected role occurs. We initially performed a screen of the major intracellular transport pathways to uncover that PGK1 acts in EGFR transport to the lysosome. We then found that the catalytic activity of PGK1 is not needed for this role, which led us to uncover that PGK1 binds directly to a dileucine motif in EGFR that has been shown previously to function as a sorting signal in promoting EGFR transport to the lysosome. Thus, these results revealed that PGK1 acts as a cargo adaptor by recognizing the dileucine sorting signal in EGFR. As PGK1 must be recruited from the cytosol to membrane for this role, we then identified two mechanisms that work

in concert to promote PGK1 recruitment to endosomal membrane. We initially find that phosphorylation of the S203 residue in PGK1 promotes its recruitment to endosomal membrane. We then identify a protein-based mechanism of recruitment, which involves the S203 phosphorylation enhancing the ability of PGK1 to interact with Hrs. We also identify a complementary lipid-based mechanism, which involves the S203 phosphorylation enhancing the ability of PGK1 to recognize PI(4,5)P$_2$ generated by PIP5K1A.

These findings advance the understanding of EGFR transport in multiple ways. First, besides the ubiquitin sorting signal that is recognized by components of the ESCRT complex, a dileucine-based sorting signal is known to be also needed for the efficient transport of EGFR to the lysosome[19,20], but the cargo adaptor that recognizes this signal has been unknown. Thus, we have advanced a mechanistic understanding of EGFR transport by revealing that PGK1 acts as a cargo adaptor that recognizes the dileucine sorting signal. Second, we have achieved a new understanding of how Hrs promotes EGFR transport to the lysosome. Besides its currently known role as an ESCRT-0 component that recognizes the ubiquitin signal on EGFR, we find that Hrs on endosomal membrane interacts with PGK1, which provides a protein-based mechanism of recruiting PGK1 to this membrane. Third,

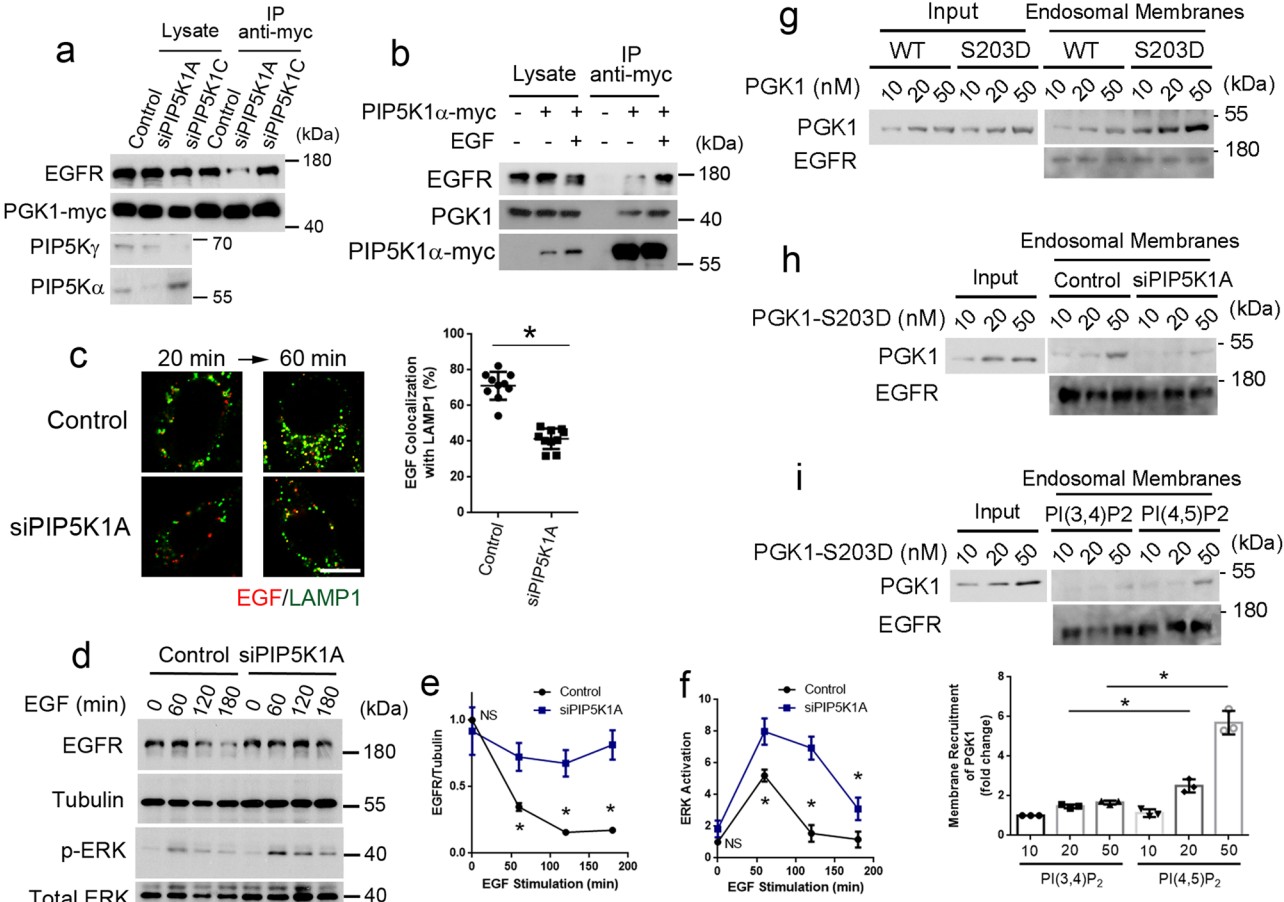

**Fig. 7 | PIP5K1A promotes PGK1 recruitment to endosomal membranes.**
Quantitative results are shown as mean +/- s.e.m.; *$p < 0.05$, unpaired two-sided
Student's $t$ test. **a** Co-precipitation analysis examining the effect of various siRNA
treatments as indicated on PGK1 association with EGFR upon EGF stimulation for
1 h, $n = 3$. **b** Co-precipitation analysis examining the effect of EGF stimulation for 1 h
on the association of PGK1 with EGFR and PIP5K1A, $n = 3$. **c** Colocalization of EGF
(red) with LAMP1 (green) upon EGF stimulation for indicated time point was per-
formed, $n = 10$ cells examined over 3 independent experiments, bar = 10 μm.
Quantitation is shown on right for a representative experiment, with statistical
analysis performed for the 60-min time point, $P = 1.29 \times 10^{-6}$. **d** Time-course analysis
of EGFR degradation upon EGF stimulation for indicated time point, examining the
effect of siRNA against PIP5K1A, $n = 3$. **e** Quantitation of EGFR level at the 0, 60, 120,
and 180-min time point for the analysis above. EGFR level was normalized to tubulin
level, *$P = 0.502$, 0.0145, 0.0085, and 0.0073, respectively. **f** Quantitation of ERK
activation at the 0, 60, 120, and 180-min time point for the analysis above.

Phosphorylated ERK level was normalized to total ERK level, *$P = 0.113$, 0.0139,
0.0121, and 0.0561, respectively. **g** Reconstitution of PGK1 recruitment to endo-
somal membrane, examining the effect of the S203D mutation on this recruitment.
Recombinant forms of PGK1 were incubated with endosomal membrane followed
by immunoblotting for PGK1 on membrane, $n = 3$. **h** Reconstitution of PGK1
recruitment to endosomal membrane, examining the effect of depleting PIP5K1A
from membrane. The S203D mutant of PGK1 was incubated with endosomal
membrane followed by immunoblotting for PGK1 on membrane, $n = 3$.
**i** Reconstitution of PGK1 recruitment to endosomal membrane, examining the
effect of delivering PI(4,5)P₂ versus PI(3,4)P₂ to endosomal membrane that was
depleted of PIP5K1A. The S203D mutant of PGK1 was incubated with endosomal
membrane followed by immunoblotting for PGK1 on membrane, $n = 3$. Quantita-
tion of PGK1 membrane recruitment by normalizing to the level of EGFR from three
experiments is shown. *$P = 0.04003$ and 0.0097 for 20 and 50 nM group compar-
isons, respectively.

we note that PI(4,5)P₂ on endosomal membrane has been found pre-
viously to promote EGFR transport to the lysosome, which involves
PI(4,5)P₂ generated by PIP5K1C being recognized by Snx5[27–29]. Advan-
cing a new understanding of how PI(4,5)P₂ promotes EGFR transport,
we find that PI(4,5)P₂ generated by PIP5K1A is recognized instead by
PGK1, and this recognition underlies a lipid-based mechanism of
recruiting PGK1 to endosomal membrane.

Coat complexes couple two major functions in mediating intra-
cellular transport, bending membrane to generate transport carriers
and binding to cargoes for sorting into these carriers. The ESCRT
complex acts as a coat complex in endosomal transport to the lyso-
some, as some of its components bind to cargoes while others bend
membrane. We have found that PGK1 binds to EGFR to promote its
transport to the lysosome. We also find that this role of PGK1 depends
on its binding to Hrs, which is a component of the ESCRT complex.
Future studies will be needed to determine more precisely whether

PGK1 binds first to Hrs and then to EGFR, or whether PGK1 engages
both Hrs and EGFR simultaneously.

In any case, our results has further revealed that PGK1 pro-
motes EGFR transport at two stages of its itinerary. One stage is
suggested by our EM finding that siRNA against PGK1 reduces the
total level of EGFR at the late endosome. Thus, when coupled with
other findings that PGK1 associates with EGFR after 15 min of EGF
stimulation and EGFR localization peaks at the early endosome at
this time point, PGK1 likely acts at the early endosome to promote
EGFR transport to the late endosome. A second stage is suggested
by quantitative EM that siRNA against PGK1 affects the ratio of EGFR
levels between the internal vesicles and the outer membrane of
the late endosome, with the siRNA treatment reducing this ratio.
This finding suggests that PGK1 also promotes the sorting of EGFR
from the outer membrane to the internal vesicles of the late
endosome.

A broader implication is also suggested by our study. Because EGFR has been intensely investigated, it has been a model for advancing a fundamental understanding of how surface proteins are downregulated through endocytic transport to the lysosome[1,3,5–7,14]. As our characterization of how PGK1 promotes EGFR transport has uncovered a sorting mechanism that complements the currently well-known mechanism that involves ESCRT recognizing the ubiquitin sorting signal, an intriguing prospect is that other endocytic proteins would also require both ubiquitin-dependent and ubiquitin-independent mechanisms working in concert to achieve efficient targeting to the lysosome.

## Methods

### Chemicals and proteins
Protein A/G agarose beads were obtained from Santa Cruz. EGFR peptides (wild-type: PNQALLRILKETE, mutant: PNQAAARILKETE) were synthesized by our core facility (https://www.ibc.sinica.edu.tw/facilities/synthesis/). EGF, Alexa 546-conjugated transferrin, Alexa 555-conjugated EGF, dextran and CTxB were obtained from Invitrogen. MEK inhibitor U1026 was obtained from Sigma. SDF-1α was obtained from R&D Systems. GST- and His-tagged fusion proteins were purified as the following statements[30]. Briefly, *Escherichia coli* (*E. coli*) strain BL21 (Novagen) was transformed with plasmids including pET32a-PGK1 (T378P, S203A or S203D), pGEX4T-1, and pGEX4T-1 containing various EGFR fragments. After induction with $500\,\mu M$ isopropyl-β-dthiogalactoside (IPTG) at $16\,^\circ C$ for 8 h, GST fusion proteins or His-tagged proteins were purified from *E. coli* lysates using glutathione-Sepharose 4B (GE Healthcare #17075605) or nickel affinity resin (Thermo #88222), respectively, according to the manufacturer's instructions. In brief, cells expressing His-tagged PGK1 were resuspended in lysis buffer ($50\,mM$ $NaH_2PO_4$ pH 8.0, $300\,mM$ NaCl, 1% Triton X-100, and protease inhibitor) followed by lysis using sonication. After binding with Ni-NTA resin, proteins were eluted with buffer containing $300\,mM$ imidazole and then dialysed using storage buffer ($25\,mM$ Tris pH 7.5 and $100\,mM$ NaCl). For GST-tagged EGFR fragments, cells were resuspended in phosphate buffered saline (PBS) with 0.5% Triton X-100, and protease inhibitor followed by binding with glutathione-Sepharose beads.

### Antibodies
The following antibodies were obtained from commercial sources: anti-Myc (Cell Signaling; 2276, immunofluorescence, 1:100/western blot, 1:1000), anti-6×His (Santa Cruz; sc-803, western blot, 1:200), anti-PGK1 (Santa Cruz; sc-130335, immunofluorescence, 1:100/western blot, 1:1000), anti-phosphoPGK1 S203 (Signalway Antibody; SAB487P, western blot, 1:500), anti-β-actin (Santa Cruz; sc-47778, western blot, 1:1000), anti-EGFR (Cell Signaling; 4267, immunofluorescence, 1:100/ western blot, 1:1000), anti-Rab11 (Cell Signaling; 5589, immunofluorescence, 1:100), anti-TfR (Santa Cruz; sc-65882, western blot, 1:500), anti-LAMP1 (Cell Signaling; 9901, immunofluorescence, 1:100/ western blot, 1:1000), anti-LAMP2 (Santa Cruz; sc-18,822, immunofluorescence, 1:100/western blot, 1:1000), anti-EEA1 (Cell Signaling; 2411, immunofluorescence, 1:100/western blot, 1:1000), anti-VAMP3 (Santa Cruz; sc-514843, western blot, 1:500), anti-Rab7 (Cell Signaling; 9367, immunofluorescence, 1:100), anti-ERK (Cell Signaling; 9107, western blot, 1:1000), anti-pERK (Cell Signaling; 4370, western blot, 1:2000), anti-PIP5K1A (GeneTex; GTX111953, western blot, 1:1000), anti-PIP5K1C (GeneTex; GTX105607, western blot, 1:1000) anti-CXCR4 (abcam; 124824, western blot, 1:1000), anti-Hrs (GeneTex; GTX101718, immunofluorescence, 1:100/western blot, 1:1000), anti-HD-PTP (Santa Cruz; sc-398711, western blot, 1:500), anti-CHMP4B (GeneTex; GTX64853, western blot, 1:1000). The following conjugated secondary antibodies were obtained from Jackson ImmunoResearch: horseradish peroxidase-conjugated donkey antibodies against mouse IgG (715-035-150, western blot, 1:10,000) and against rabbit IgG (711-035-152,

western blot, 1:10,000), Cy2 donkey antibodies against mouse IgG (715-225-151, immunofluorescence, 1:200) and against rabbit IgG (711-225-152, immunofluorescence, 1:200), Cy3 goat antibody against mouse IgG (115-165-062, immunofluorescence, 1:200) and Cy3 donkey antibodies against rabbit IgG (711-165-152, immunofluorescence, 1:200). DAPI (Sigma) was used for nuclear staining.

### Cell culture and EGF treatment
HeLa cells (#CCL-2) from ATCC were cultured in Dulbecco's Modified Eagle Medium (DMEM) with 10% fetal bovine serum (FBS) and supplemented with glutamine and penicillin/streptomycin. A549 cells (#CCL-185) from ATCC were cultured in Ham's F-12K (Kaighn's) Medium with 10% fetal bovine serum (FBS) and penicillin/streptomycin. DNA plasmids were transfected using FuGene6 (Roche). Oligonucleotides for siRNA experiments were transfected using PepMute (SignaGen). For EGF treatment, cells were serum-starved, and then treated with EGF (100 ng/ml) for 1 h at $37\,^\circ C$, unless indicated otherwise in the figure legend.

### Sequences for siRNA
Sequences for siRNA against PGK1, CACAAGCUGGACAGCCAUG (siPGK#1) and GCUUCUGGGAACAAGGUUA (siPGK#2), and also for siRNA against PIP5K1A, GGUGCCAUCCAGUUAGGCA, were obtained from Dharmacon. Sequences for siRNA against ESCRT complex and PIP5K1C, GACAACGACUUCAUUUACC (siHD-PTP), CGUCUUUCCAGA AUUCAAA (siHrs), CCAGUCUUCUCUCGUCCUA (siTSG101), GGAU GGGAGGUACUGGAUU (siPIP5K1C), and CAUCGAGUUCCAGCGGGAG (siCHMP4B) were obtained from BioTools. Rescue plasmids for expression of PGK1 were generated using QuikChange Site-Directed-Mutagenesis (Stratagene).

### Plasmids
PGK1 forms, wild-type, catalytic dead (T378P), and phosphorylation mutants (S203A and S203D) were cloned into the BamHI and XhoI sites of the pcDNA3.1-myc-His(-), pET32a (Invitrogen), and pGEX4T-1 (Amersham) vectors. PIP5K1A forms, wild-type and catalytic dead (D309N R427Q) were also cloned into the BamHI and XhoI sites of the pcDNA3.1-myc-His(-) vector. To append the cytoplasmic domain of EGFR to the carboxy terminus of GST, the cDNA encoding different domains or fragments was amplified by PCR and then subcloned into the BamH1 and XhoI sites of pGEX-4T-1 vector (Amersham).

### Pull-down assays using purified proteins
GST-fusion proteins on glutathione beads were incubated with soluble proteins (100 nM) at $4\,^\circ C$ for 1 h in incubation buffer (PBS with 0.05% Triton X-100 and protease inhibitors). Beads were then collected by centrifugation at 800 x $g$ for 3 min at $4\,^\circ C$, followed by two washes with incubation buffer. Samples were then analyzed using SDS-PAGE followed by either Coomassie staining or western blot.

### Co-precipitations using cell lysates
For immunoprecipitation of Myc-tagged proteins, HeLa cells expressing Myc-tagged proteins were disrupted in lysis buffer (PBS with 0.5% Triton X-100 and protease inhibitors). Lysates were cleared by centrifugation at 13,000 x $g$ for 15 min at $4\,^\circ C$, and were then incubated with anti-Myc antibodies for 2 h at $4\,^\circ C$ followed by incubation with protein A beads (Santa Cruz). The beads were then washed three times with lysis buffer and then analyzed using western blot.

### Cell-based transport assays
A quantitative microscopy-based assay, which involves the colocalization of model cargoes with organelle markers and coupled with kinetic analysis, was performed as previously described[23,30,31].

Briefly, to examine anterograde transport from ER to Golgi, cells were transfected with pROSE-VSVG-ts045-Myc for 1 day, and then

incubated at 39 °C for 4 h to accumulate VSVG in the ER. Cells were then shifted to 32 °C for different times as indicated in figures. Confocal microscopy was then performed to assess the colocalization of VSVG with giantin, a *cis*-Golgi marker, which tracks VSVG arrival to the Golgi.

To examine retrograde transport from the Golgi to the ER, cells were transfected with pROSE-VSVG-ts045-KDELR-Myc for 1 day, and then incubated at 32 °C for 8 h to achieve steady-state distribution at the Golgi. Cells were then shifted to 39 °C for different times as indicated in the figures. Confocal microscopy was then performed to assess the colocalization of VSVG-KDELR with giantin, which tracks exit of VSVG-KDELR from the Golgi.

To examine anterograde transport from the Golgi to the PM, cells were transfected with pROSE-VSVG-ts045-Myc for 1 day, and then incubated at 20 °C for 2 h to accumulate VSVG at the TGN. Cells were then shifted to 32 °C for different times as indicated in the figures. Confocal microscopy was then performed to assess the colocalization of VSVG with TGN46, a TGN marker, which tracks the exit of VSVG from the Golgi.

To examine retrograde transport from the PM to the Golgi, cells were incubated with Alexa 555-conjugated CTxB (0.5 μg/ml in DMEM) for 30 min at 4 °C. After washing to release unbound CT, cells were shifted to 37 °C for different times as indicated in the figures. Confocal microscopy was then performed to assess the colocalization of CTxB with TGN46, which tracks the arrival of CTxB to the Golgi.

To examine endocytosis, cells were serum-starved for 1 h and further incubated with Alexa 555-conjugated EGF (100 ng/ml in DMEM) for 1 h at 4 °C. Cells were then washed to clear unbound EGF, followed by shifting to 37 °C for times indicated in the figures. Confocal microscopy was then performed to assess the colocalization of EGF with EEA1, an early endosome marker, which tracks the arrival of EGF to the early endosome.

To examine endocytic recycling, cells were incubated with Alexa 546-conjugated transferrin (Tf) (5 μg/ml in DMEM) at 37 °C for 2 h to accumulate Tf in endosomes. Cells were then incubated with medium without Tf for different time points as indicated in the figures. Confocal microscopy was then performed to assess the colocalization of Tf with Rab11, a recycling endosome marker, which tracks the exit of Tf from the recycling endosome.

To examine endocytic transport to the lysosome, cells were serum-starved for 1 h and then incubated with Alexa 555-conjugated EGF (100 ng/ml in DMEM) for 1 h at 4 °C. Cells were then washed to release unbound EGF, followed by shifting to 37 °C for times indicated in the figures. Confocal microscopy was then performed to assess the colocalization of EGF with Lamp1, a lysosome marker, which tracks the arrival of EGF to the lysosome.

Quantitation of these transport assays involves three independent experiments. For each experiment, we examined 10 cells for each time point, and statistics was performed for each of these time points.

### CXCR4 degradation assay
HeLa cells were initially incubated in DMEM complete media with 10% fetal bovine serum containing 10 μg/ml cycloheximide for 1 h at 37 °C. Cells were then incubated in the same media in the presence of 100 nM stromal derived factor-1α (SDF-1α) for 0, 1, 3, or 6 h of the incubation. Cells were collected and lysed in the lysis buffer containing 50 mM Tris pH 7.4, 150 mM NaCl, 5 mM EDTA and 1% Triton X-100. The cell lysates were then analyzed using SDS-PAGE followed by western blot.

### EGFR degradation assay
Degradation of EGFR was measured essentially as previously described[32]. In brief, serum-starved cells were pre-treated with 10 μg/mL cycloheximide for 1 h and then chased with 100 ng/ml EGF at 37 °C for different time points as indicated in the figures. At the end of the chase time, cells were collected and lysed in the lysis buffer containing 50 mM

Tris pH 7.4, 150 mM NaCl, 5 mM EDTA and 1% Triton X-100. The cell lysates were then analyzed using SDS-PAGE followed by western blot.

### TfR degradation assay
Degradation of TfR was measured as previously described[25]. Briefly, serum-starved cells were pre-treated with 10 μg/mL cycloheximide and then chased with 100 μg/ml FAC at 37 °C for different time points as indicated in the figures. At the end of the chase time, cells were collected and lysed in the lysis buffer containing 50 mM Tris pH 7.4, 150 mM NaCl, 5 mM EDTA and 1% Triton X-100. The cell lysates were then analyzed using SDS-PAGE followed by western blot.

### Subcellular fractionation
HeLa cells were resuspended in homogenization buffer containing 20 mM HEPES-KOH, pH 7.4, 250 mM sucrose, 1 mM EDTA, and protease inhibitors and then lysed by passing through a 25-gauge needle to obtain cell homogenates. After centrifugation at 800 x $g$ for 5 min to pellet unbroken cells, the cleared lysates were further centrifuged at 13,000 x $g$ for 10 min to separate heavy membranes. The supernatant was next subjected to velocity centrifugation at 100,000 x $g$ for 1 h to obtain light membranes that contain endosomes versus cytosol. Equal fractional amounts were then analyzed by SDS-PAGE followed by western blot.

### Proximity ligation assay
PLA was conducted using a PLA kit (Duolink® In Situ, Sigma) according to the manufacturer's instructions. Antibodies used for this assay were as follows; rabbit anti-EGFR (Cell Signaling; 4267), rabbit anti-PGK1 (Cell Signaling; 68540), and mouse anti-Myc (Cell Signaling; 2276). Nuclear staining was done using mounting medium with DAPI.

### ATP detection assay
ATP level in cells was detected using ATPlite Luminescence Assay System (PerkinElmer; 6016943) according to the manufacturer's instructions with final values normalized for cell number. Luminescence was measured with a multimode microplate reader (BioTek Synergy H1).

### Isolation of endosomal membrane
HeLa cells were collected and resuspended in homogenization buffer containing 320 mM sucrose and 25 mM Tris pH 7.4. Cells were homogenized using a Dounce homogenizer, and were then centrifuged at 2000 x $g$ for 10 min. The resulting supernatant was loaded onto the top of a sucrose gradient containing 29% sucrose (top) and 35% sucrose (bottom). After centrifugation at 110,000 x $g$ for 2.5 h, the endosomal membrane fraction was collected at the 29% and 35% sucrose interface.

### Reconstitution of PGK1 recruitment to endosomal membrane
Isolated endosomal membrane (0.1 mg/ml) in 500 μl of traffic buffer (25 mM HEPES pH 7.2, 50 mM KCl, 2.5 mM Mg(OAc)₂, and 200 mM sucrose) was washed and then incubated for 15 min at 37 °C with 100 nM of recombinant PGK1. Samples were centrifuged for 10 min at 13,000 x $g$ to recover endosomal membrane in the pellet fraction. The supernatant fraction was concentrated by trichloroacetic acid (TCA) precipitation. Equal fractional amounts were then analyzed by western blotting.

To reconstitute PGK1 recruitment by delivering a specific lipid to endosomal membranes that had been depleted of PIP5K1A, cells were treated with siRNA against PIP5K1A and then fed with liposomes [125 μM PC-PE-PI(4,5)P₂ or PC-PE-PI(3,4)P₂] that were generated in the presence of 25 μM bovine serum albumin for 3 h. Endosomal membranes were then isolated from these cells for the reconstitution of PGK1 recruitment.

## Quantification of PI(4,5)P2 and PI(3,4)P2 on endosomal membrane

Lipids were extracted from endosomal membrane fractions, and then PI(4,5)P2 and PI(3,4)P2 levels were quantified by a competition ELISA approach, using PI(4,5)P2 Mass ELISA (K-4500) and PI(3,4)P2 Mass ELISA (K-3800) kits from Echelon Biosciences, as described by the manufacturer's instructions.

## Confocal microscopy

Colocalization studies were performed using the Zeiss LSM 900 Confocal Microscope. For quantitation of colocalization, ten fields of cells were examined, with each field typically containing 5 cells. Images were imported into the NIH Image J version 1.50i software. Under the "Image" tab, the "Split Channels" option was selected. Under the "Plugins" tab, "Colocalization Analysis" option was selected, and within this option, the "Colocalization Threshold" option was selected. The threshold values were chosen automatically by the program and Manders' coefficients were then calculated and expressed as the fraction of protein of interest colocalized with an organelle marker. All confocal microscopy images were cropped to proper sizes by Zeiss LSM 900 acquisition software installed with the confocal system.

## Electron microscopy

EGFR level in internal vesicles of late endosome was quantified by electron microscopy (EM) as previously described[32]. Cells treated with siRNA against PGK1 or not were starved overnight, incubated with anti-EGFR antibody (Millipore, LA22) at 4 °C for 30 min, washed and followed by incubation at 4 °C for additional 30 min with 15 nm protein A-gold (Electron Microscopy Sciences). Cells were then washed, stimulated with 100 ng/ml EGF for 1 h at 37 °C, fixed with 2% glutaraldehyde in 0.1 M sodium phosphate buffer and processed for EM examination in the EM facility. The ultrasections were viewed using a Jeol JEM-1400 transmission electron microscope.

## Statistical analysis

Quantitative data are shown as mean with standard error. Statistical significance was determined using the two-tailed Student's $t$ test (through Excel or Prism software). For figures, $*P < 0.05$ or NS (non-significant) $P > 0.05$ is shown. In the accompanying figure legend, the exact values are shown.

## Reporting summary

Further information on research design is available in the Nature Portfolio Reporting Summary linked to this article.

## Data availability

Source data including all raw data generated in this study are provided with this paper. The following figures have associated raw data listed in the source data file: Figs. 1a–j, 2b, c, e–h, 3d, f, g, 4f, g, i, j, 5e, 6a, b, d, e, 7c, e, f, i and Supplementary Figs. 1j, 2a–c, f, g, 3b–d, f, 4b, f–m, 5a–c, 6a–c, e, 7a–d, 8a, c–e, g–j. All data supporting the findings of this study are available from the corresponding authors upon request. Source data are provided with this paper.

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

## Acknowledgements

We thank Fang-Jen S. Lee and Peter Chi for providing cell lines and Ping-Hung Chen for providing EGFR plasmid. We also thank Po-Huang Liang for sharing reagents. We thank Joint Center for Instruments and Research, College of Bioresources and Agriculture, National Taiwan University for the technical support of electron microscopy. This work is supported by grants from the National Science and Technology Council (NSTC), Taiwan (109-2636-B-002-015, 110-2636-B-002-012, 111-2636-B-002-029, and 112-2636-B-002-006) to J.-W.H., the Yushan Fellow Program by the Ministry of Education (MOE), Taiwan to J.-W.H., and the National Institute of Health, USA to V.W.H. (R37GM058615).

## Author contributions

SLC performed the majority of experiments including transport assays, colocalization analysis, and protein-protein interaction assays. JWH performed transport assays and PLA assays. YTC conducted EGFR degradation and ERK activity assays. JRH and SYY performed the dileucine-related studies and supported other biological experiments. JSY performed ER and Golgi transport assays. SLC, JWH, and VWH designed the study and analyzed the data. JWH supervised the study with help from VWH. JWH and VWH wrote the manuscript with input from all other authors.

## Competing interests

The authors declare no competing interests.
