## [Peer Review File · Nature Communications]

Phosphoglycerate kinase 1 acts as a cargo adaptor to promote EGFR transport to the lysosomeREVIEWER COMMENTS

Reviewer #1 (Remarks to the Author):

Chu et al. described a novel role of PGK1 in the regulation of EGFR degradation and signaling. In particular, they showed that PGK1 directly binds to the EGFR, that this association is increased upon EGF stimulation and is responsible for EGFR targeting to the lysosome. Indeed, PGK1 KD delayed EGFR trafficking to the lysosome and its degradation, while it caused the sustaining of downstream signaling. They also investigated the molecular mechanism at the basis of this recruitment showing that it occurs through the di-leucine motif in the EGFR juxta-membrane region. The authors also proposed that PIP2 and Hrs both contribute to PGK1 recruitment to the endosomal membrane.

The findings herein described are novel and of great interest for cell biology, trafficking and signaling community, as they elucidate a novel mechanism involved in the transport of EGFR to the lysosome, which does not depend on ubiquitin but on PGK1 and Hrs (possibly independently on receptor ubiquitination but dependent on sorting signals in the EGFR tail).

While the work went deep in the understanding of the molecular mechanism of PGK1 recruitment, there is no deep investigation nor discussion about the link between the known and well-characterized function of PGK1 in glycolysis with the one in EGFR transport. Are the two functions connected or totally independent? For instance, given that the catalytically inactive form of PGK1 seems to rescue the degradation defect in the PGK1-KD cells, this would suggest two independent functions. However, this is not discussed. I would suggest to add more contextualization about the known previous functions of PGK1 and some discussion of the possible link between EGFR trafficking and cell metabolism.

On the experimental side, there are a number of technical issues that need to be solved, listed below.

1) Lysosomal targeting of the EGFR is induced by EGF and indeed EGFR-PGK1 interaction is increased upon EGF. However, in some experiments, authors stimulate cells with EGF while in others do not, and this is not explained nor rationalized. For instance, all interactions with late endosomal ESCRT components in Figure 5 are apparently done in absence of EGF and this should be repeated in presence of EGF as this might affect the result, given that EGFR is relocalized at late endosomes upon EGF. The same applies to Fig. 3J. These experiments should be repeated upon EGF stimulation.

Related to this, it is very important that authors state in the figure if experiments are performed in the presence of EGF and at which time points (see also comments below), otherwise it is very difficult for the reader to understand how experiments are performed.

2) Many experiments lack critical controls.

- Fig. 2h: Authors should analyze in the same experiments Control, siPGK1 and the reconstitution of siPGK1 with PGK1 WT and mutant. This is critical to show that PGK1 WT and the mutant are both indeed rescuing the trafficking defect observed upon PGK1 KD.

- Fig.4d-K and Fig. 6c-e: all experiments lack the WT sample for reference. It's not possible to conclude that the S203A mutant is defective, and even less possible to conclude that the S203D mutant is increasing the phenotype if it is not compared with PGK1 WT in the same experiment. Thus, to make conclusions, authors need to repeat experiments together with WT.

- Experiments with endosomal membrane purification lack controls for good purification, e.g., enrichment in endosomal components and depletion of other membranes.

- Experiments with soluble/pellet fraction in Fig. 4a, b, d lack the INPUT/TOTAL control.

3) The EGFR-LA mutant in Figure 3D is not convincing. It seems that this mutant is diffused in the cytosol, possibly because it is defective in being exported to the PM. Authors should provide evidence that this mutant is correctly localized at the PM and is not trapped in the biosynthetic pathway. They should provide also staining of the mutant and the WT in basal condition (without EGF stimulation), to show the localization of the mutant. In addition, they should provide a trafficking assay, e.g., with labeled EGF at different time points at 37, or left for 1 h at 4C to see the amount of total EGFR at the PM for the WT and the mutant.

4) The phenotype of the siPGK1 on EGFR colocalization with Lamp1 (Fig. 1g) is more a delay than a decrease in colocalization. Similarly, the effect of the PGK1 overexpression (Fig. 1h) is an acceleration and not an increase. This should be revised in the text.

Figures:

In Fig. 1, Fig. 3F, Fig. 4h, Fig. 7e and 7f: quantifications and statistics should be shown for all the time points of the degradation/signaling curve.

Related to all figures, authors should revise all the nomenclature to render them clearer:

1) Add indication and time of EGF stimulation

2) Provide details in the figure about which mutant is used in the experiment. Authors generated many mutants, e.g., of the EGFR, PGK1, and mutant peptides... and they could not refer to them simply as "mutant" in the figures but they have to specify which one they are (see, for instance, Fig. 3d, e, f, g, I, J and others...)

3) Fig. 7 g, h, i, please specify in all cases in the figure that experiments are performed with purified endosomal membranes.

4) Figure legends need to be revised adding more information in order for the reader to more easily understand what is shown.

Reviewer #2 (Remarks to the Author):

In this paper, authors report that phosphorylated phosphoglycerate kinase 1 (PGK1) is involved in EGFR transport to lysosomes for degradation, that PGK1 binds a diLeu motif that had been previously identified in EGFR cytoplasmic domain and that PGK1 S203 phosphorylation stimulates EGFR degradation without affecting EGFR-PGK1 association. They also report that the recruitment of PGK1 to endosomes depends on PIP2 and Hrs. Authors conclude that their study reveals the existence of an unexpected coat function for a metabolic enzyme, and improves our knowledge of EGFR transport to lysosomes. The topic of the paper is clearly interesting since little is known about the direct role of metabolic enzymes in trafficking, in particular in the downregulation of EGFR. However, some serious shortcomings dampen my enthusiasm for this study.

Main comments

1) The observations that PGK1 binds EGFR and that this binding may depend on the presence of an intact diLeu motif in EGFR tail does not demonstrate that PGK1 is a coat adaptor. This is a serious overstatement. What is the coat, and which sorting step does it mediate?

Does PGK1 binding to EGFR (diLeu motif) and to Hrs (regulated by PGK1 S203 phosphorylation, Fig 6c-d) correspond to the same sorting step? Or to sequential steps? Authors argue that PGK1 S203 phosphorylation targets PGK1 primarily to late endosomes (ED Fig 4). Does it mean that PGK1 binds EGFR in late endosomes? What is the relationship between these mechanisms?

2) Information in text and legends is often missing or so limited that it is not easy to understand how certain experiments were performed. This is not acceptable. Some examples:

- it is often not clear whether cells were treated with EGF, and if so at what concentration and for how long (eg Fig 3a and 6). Is EGFR endocytosis (Fig 1g-h) always elicited after treatment with 100 ng/ml EGF for 1h at 4°?

- The Legend of each panel of Fig 1a-j says: "Statistics is shown for a representative experiment examining the x-min time-point"? Does it mean that error bars are shown for n=1? The same comment applies elsewhere.

- How were PLA experiments done? For example, in ED Fig 2, were cells treated with EGF? What are the PLA puncta? Were cells labeled with antibodies to EGFR or to myc tag? Similarly, it is not possible to evaluate the PLA analysis in Fig 4f, 5e and 6d.

- What tag is used for the wt and mutant form of PGK1 in Fig ED 2b? Also, the legends and text provide no information on how PGK1 is labeled in ED Fig 4. Do the panels show endogenous PGK1 labeled with

antibodies? Or tagged, overexpressed PGK1? How are EEA1, RAB7 etc.. labeled? Are these confocal images?

- In the reconstitution experiments in Fig 7i, cellular membranes prepared by subcellular fractionation were incubated with liposomes containing PIP2 or PI(3,4)P2 to modulate the phosphoinositide content of the membranes. How was PIP2 or PI(3,4)P2 delivered to membranes? By liposome fusion? If so, how was this done? How were liposomes then separated from membranes so that PGK1 recruitment could be studied?

3) Is PGK1 really present on endosomes / lysosomes containing EGFR? Unfortunately, the analysis of PGK1 subcellular distribution by immunofluorescence is not convincing. In ED Fig 4 and all other figure, PGK1 shows a reticular, grainy pattern across the cytoplasm (excluding the nucleus), reminiscent of cytosol labeling or of background (does PGK1 staining disappear after PGK1 KD?). Moreover, this PGK1 labeling pattern is clearly different from the punctate patterns of EEA1, RAB7, LAMP1 or Hrs. How can PGK1 show 50-80% colocalization with these markers?? This is simply not possible, and colocalization of PGK1 with any of these markers seems fortuitous. The same comments apply to ED Fig 5 and ED Fig 6a, b, c and Fig 6b and 6e.

4) The experiments that monitor TfR degradation in the presence of ferric ammonium citrate are not convincing (Fig 1j), given the very long time-course of these experiments (24h). Since protein synthesis goes on, these experiments monitor the total cellular levels of TfR over 24h with/without ferric ammonium citrate, and not TfR transport to the lysosomes for degradation.

5) The immuno-EM data in ED Fig 5d are not convincing. It is not possible to distinguish intraluminal vesicles in these micrographs. Neither is it clear what are the gold particles. Finally, the quantification shows that the total number of anti-EGFR gold particles is reduced after PGK1 KD and not the distribution of gold on limiting vs. intraluminal membranes. This suggests that PGK1 is involved in EGFR transport towards MVBs and not EGFR incorporation into internal vesicles. Finally, the y-axis of panel 5D should not be EGFR per MVB but gold particles per profile.

Other comments

1) Authors mention that cargo proteins are sorted into internal vesicles of the late endosome (Multivesicular Body, MVB) for targeting to lysosome or remain at its delimiting membrane for recycling to the cell surface. This is inaccurate: i) most recycling receptors are sorted in early endosomes in order to recycle to the cell surface (e.g. TfR used in the paper); ii) ESCRT-mediated sorting begins in early endosomes, and ESCRT-I to III are primarily found on early endosomes, including Hrs, which is used in the paper (see for example Wenzel Stenmark Raiborg Nat Comm 2018). This point is important for the discussion.

2) In Fig 1g, siPGK1 delays EGFR transport to LAMP1 by 20min. This is not a very strong inhibition. Also, how much is PGK1 overexpressed in Fig 1h and Fig 2d-e?

3) PGK catalyzes the first ATP-generating step in glycolysis. Are ATP levels altered after PGK1 depletion with siRNAs or PGK1 overexpression?

4) Authors argue that EGFR degradation is reduced by anti-PGK1 siRNAs (Fig 2a). If so, I do not understand why the total cellular amounts of EGFR are similar in control cells and in KD cells (Fig 2a, time 0).

5) Fig 2a shows a representative experiment of EGFR degradation in time with 2 siRNAs and n=3. Which one of the two siRNAs is used for quantification in Fig 2b-c, since each graph shows 3 data points?

6) Authors report that wt or catalytically dead (T378P) PGK1 can be expressed in cells after PGK1 KD, in order to obtain physiological levels of either protein (Fig 2b). However, as I understand it, EGFR degradation is not measured after PGK1 KD followed by re-expression, but after overexpression of wt or catalytically dead PGK1 (Fig 2h and ED Fig 2c-d). Is this correct? If so, authors cannot conclude that the catalytic activity of PGK1 is not required for EGFR transport. Please clarify.

7) It is not clear how the quantitative analysis of EGFR distribution was carried out (ED Fig 2a, ED Fig 3c-e, Fig 2h, 3d, 4g). Did authors quantify the number of structures that contain a given markers or the intensity of the marker? This is important since treatments that have been used (e.g. RNAi) may change the number of labeled structures (e.g. endosome fragmentation / coalescence).

8) The effect of the MEK inhibitor UO126 is not very convincing in Fig 4a. Also, ED Fig 4a-c indicates that the sum of the percentages of EGFR or PGK1 that colocalize with endosomal markers is greater than 100%. Authors conclude that this is due to the fact that endosomal markers exhibit overlapping distributions (ED Fig 4d). That may be, but I find it very surprising that 40% of RAB7 colocalizes with EEA1 (ED Fig 4D). A simpler interpretation is that the labeling of the two markers is clustered in the same region, and thus overlaps in the same confocal volume (even if present on different membranes).

9) The data showing that EGF stimulation promotes the interaction between PGK1 and Hrs (Fig 6a) are not really convincing. The effects are small and, without quantification, it is not possible to determine whether these effects are significant.

10) Authors state that endosomal membranes are analyzed in Fig 6F (and elsewhere in the paper). From the Methods, it is not clear whether these membranes were obtained after centrifugation at 100'000xg for 1h or after sedimentation onto a step gradient. In any case, these fractions are not endosomal membranes (Fig 6f), but relatively crude membrane fractions (e.g. do these fractions contain both early and late endosome markers?). Also, an independent equal loading control should be shown (not EGFR).

11) The quantification in Fig7c shows that approx. 70% EGFR colocalizes with LAMP1 under control conditions, but a lot of red structures (EGFR) do not colocalize with green (LAMP1).

12) What is the subcellular distribution of PIP5K1A with or without stimulation with EGF? Is PIP5K1A found on endosomes containing EGFR?

Reviewer #3 (Remarks to the Author):

The epidermal growth factor receptor (EGFR) is responsible for signalling events at the plasma membrane, in endosomes and in the nucleus that lead to modulation of vital cellular processes such as proliferation and differentiation. Therefore understanding the intracellular sorting mechanism leading to its downregulation is of crucial importance. Here, Chu et al report a novel endosomal adaptor, PGK1 that is recruited to endosomal membranes upon its phosphorylation and recognizes a dileucine motif on EGFR for targeting to the lysosomal system. The authors find that this process is mediated by PGK1 interaction with Hrs and PI(4,5)P2. The work is overall significant to the field and provides substantial evidence that supports the claims. I hope that the comments below will help improve the manuscript:

1. PGK1 is a cytosolic protein, therefore colocalization studies are difficult to interpret. The staining of PGK1 in Fig.6e and ED Fig.4 and ED Fig.5 reveals a homogeneous distribution of the protein in the cytosol and even in the nucleus in some instances. The images do not convincingly show the recruitment or an enrichment of PGK1 in endosomal structures. Perhaps this could be better inferred from the gray images of the individual channels or by employing a different assay altogether. Positive/negative controls are missing.

2. The high percentage of colocalization between PGK1 S203D mutant and Hrs in Figure 6.e is misleading as it is clear from the microscopy images that most of the protein is still in the cytosol and not in endosomes.

3. Significance tests were only performed for specific time points throughout the manuscript and indicators should mark only these, otherwise is misleading.

4. The specific time points for which quantification was performed should be mentioned in the Figures throughout the manuscript (e.g. Fig.1j, Fig.2 b,c,e,f etc).

RESPONSES (NCOMMS-22-46410-T)

We thank the reviewers for the overall positive assessment of our work. Each has also provided comments that suggest how we can further improve our study and its presentation. Taking them to heart, we have performed many additional experiments, which are shown in revised figures or in new figures. They are described below as part of the point-by-point response to all comments of the last review. Moreover, to help detect the many changes in the revised manuscript, we have highlighted the text in blue to indicate where changes are made.

Reviewer #1 (Remarks to the Author):

Chu et al. described a novel role of PGK1 in the regulation of EGFR degradation and signaling. In particular, they showed that PGK1 directly binds to the EGFR, that this association is increased upon EGF stimulation and is responsible for EGFR targeting to the lysosome. Indeed, PGK1 KD delayed EGFR trafficking to the lysosome and its degradation, while it caused the sustaining of downstream signaling. They also investigated the molecular mechanism at the basis of this recruitment showing that it occurs through the di-leucine motif in the EGFR juxta-membrane region. The authors also proposed that PIP2 and Hrs both contribute to PGK1 recruitment to the endosomal membrane.

The findings herein described are novel and of great interest for cell biology, trafficking and signaling community, as they elucidate a novel mechanism involved in the transport of EGFR to the lysosome, which does not depend on ubiquitin but on PGK1 and Hrs (possibly independently on receptor ubiquitination but dependent on sorting signals in the EGFR tail).

While the work went deep in the understanding of the molecular mechanism of PGK1 recruitment, there is no deep investigation nor discussion about the link between the known and well-characterized function of PGK1 in glycolysis with the one in EGFR transport. Are the two functions connected or totally independent? For instance, given that the catalytically inactive form of PGK1 seems to rescue the degradation defect in the PGK1-KD cells, this would suggest two independent functions. However, this is not discussed. I would suggest to add more contextualization about the known previous functions of PGK1 and some discussion of the possible link between EGFR trafficking and cell metabolism.

We have now revised the manuscript to say that the role of PGK1 in EGFR transport is likely to be independent of its canonical role in glycolysis, as the catalytic activity of PGK1 is not needed for its role in EGFR transport. We have also performed two additional experiments that further support this conclusion. First, we find that glucose starvation, which should reduce PGK1 function in glycolysis, still allows EGF stimulation to induce PGK1 to associate with EGFR (new ED Fig 3e). Second, altering the cellular level of PGK1, either by reducing its level through siRNA treatment or increasing its level through overexpression, has a modest effect on the cellular ATP level (new ED Fig 3f), in contrast to the more appreciable effects that these perturbations have on EGFR transport. This difference could be explained by the coat function of PGK1 predicted to require stoichiometric protein level, while its glycolytic function would only need catalytic amount.

These above statements have been incorporated into the Result section that refer to the new figures.

1) Lysosomal targeting of the EGFR is induced by EGF and indeed EGFR-PGK1 interaction is increased upon EGF. However, in some experiments, authors stimulate cells with EGF while in others do not, and this is not explained nor rationalized. For instance, all interactions with late endosomal ESCRT components in Figure 5 are apparently done in absence of EGF and this should be repeated in presence of EGF as this might affect the result, given that EGFR is relocalized at late endosomes upon EGF. The same applies to Fig. 3J. These experiments should be repeated upon EGF stimulation.

Related to this, it is very important that authors state in the figure if experiments are performed in the presence of EGF and at which time points (see also comments below), otherwise it is very difficult for the reader to understand how experiments are performed.

We apologize for the confusion. As EGFR transport to the lysosome requires EGF stimulation, and we showed in initial figures that EGF stimulation recruits PGK1 to endosomal membrane and induces PGK1 association with EGFR, we then skipped the no (EGF) stimulation condition in subsequent figures due to space limitation. We have now revised the figure legends to state specifically whether EGF stimulation was done. We have also repeated Fig 3j and Fig 5e to show as examples the difference between EGF stimulation vs no stimulation (see revised Fig 3j and Fig 5e).

2) Many experiments lack critical controls.

- Fig. 2h: Authors should analyze in the same experiments Control, siPGK1 and the reconstitution of siPGK1 with PGK1 WT and mutant. This is critical to show that PGK1 WT and the mutant are both indeed rescuing the trafficking defect observed upon PGK1 KD.

As requested, we have repeated Fig. 2h with the additional conditions of control and siPGK1 in order to show more explicitly that the expression of PGK1-WT or PGK1-T378P in siPGK1 cells rescues the defect in EGFR transport induced by si-PGK1 (see revised Fig 2h).

- Fig.4d-K and Fig. 6c-e: all experiments lack the WT sample for reference. It's not possible to conclude that the S203A mutant is defective, and even less possible to conclude that the S203D mutant is increasing the phenotype if it is not compared with PGK1 WT in the same experiment. Thus, to make conclusions, authors need to repeat experiments together with WT.

As requested, we have added the wild-type PGK1 condition for comparison (see revised Figs 4d, 4e, 4f, 4g, 4h, 4i, 4j, 6c, 6d, 6e). Due to space limitation, the results are shown for the EGF stimulation condition, and we have stated as such in the revised figure legends. Note that Fig 4k is an in vitro experiment using recombinant PGK1 forms, and thus WT would be similar to S203A.

- Experiments with endosomal membrane purification lack controls for good purification, e.g., enrichment in endosomal components and depletion of other membranes.

As requested, we now show that the enriched endosomal membrane fraction contains EEA1 (early endosome marker) and CD63 (late endosome marker), but not calnexin (ER marker) and COX4 (mitochondria marker) (new ED Fig 6d).

- Experiments with soluble/pellet fraction in Fig. 4a, b, d lack the INPUT/TOTAL control.

As requested, we have added the input amounts (see revised Fig. 4a, 4b, 4d).

3) The EGFR-LA mutant in Figure 3D is not convincing. It seems that this mutant is diffused in the cytosol, possibly because it is defective in being exported to the PM. Authors should provide evidence that this mutant is correctly localized at the PM and is not trapped in the biosynthetic pathway. They should provide also staining of the mutant and the WT in basal condition (without EGF stimulation), to show the localization of the mutant. In addition, they should provide a trafficking assay, e.g., with labeled EGF at different time points at 37, or left for 1 h at 4C to see the amount of total EGFR at the PM for the WT and the mutant.

We apologize for the poor quality of the image for the EGFR-LA mutant in Fig 3d, as it is overexposed, leading to the impression that the mutant is cytosolic. We have now provided a more appropriately exposed image, which shows the mutant to have a more punctate pattern, consistent with its transport through the endosomes (see revised Fig 3d). We also wish to point out that this mutant has already been documented previously to be transported to the PM (PMID: 10942518). Nevertheless, we have further confirmed this with an additional experiment. In contrast to microscopy-based studies, the biochemical approach of biotinylation can more readily appreciate the level of proteins on the PM. Taking this approach, we detected the surface pool of WT and mutant EGFR by labeling cells with biotin-labeled EGF at 4°C and find that their levels are similar (new ED Fig 3c). Moreover, with respect to tracking the endocytic fate of the mutant upon EGF stimulation, we have performed the following additional experiment. We incubated cells with fluorescence-labeled EGF at 40C to label the surface pool of the mutant EGFR. We then tracked its endosomal transport upon EGF stimulation (new ED Fig 3b).

4) The phenotype of the siPGK1 on EGFR colocalization with Lamp1 (Fig. 1g) is more a delay than a decrease in colocalization. Similarly, the effect of the PGK1 overexpression (Fig. 1h) is an acceleration and not an increase. This should be revised in the text.

We have revised the text as requested.

Figures:

In Fig. 1, Fig. 3F, Fig. 4h, Fig. 7e and 7f: quantifications and statistics should be shown for all the time points of the degradation/signaling curve.

We have now done so (see revised Fig 1, 3f/g, 4i/j, 7e, 7f).

Related to all figures, authors should revise all the nomenclature to render them clearer:

1) Add indication and time of EGF stimulation

2) Provide details in the figure about which mutant is used in the experiment. Authors generated many mutants, e.g., of the EGFR, PGK1, and mutant peptides... and they could not refer to them simply as “mutant” in the figures but they have to specify which one they are (see, for instance, Fig. 3d, e, f, g, I, J and others...)

- 3) Fig. 7 g, h, i, please specify in all cases in the figure that experiments are performed with purified endosomal membranes.
- 4) Figure legends need to be revised adding more information in order for the reader to more easily understand what is shown.

We have now revised many figure legends to address the comments above, specifically:

- 1) We have added the time of EGF stimulation when it is indicated.
- 2) We have specified what mutants are used.
- 3) We have labeled that endosomal membranes were used in Fig 7g, 7h, 7i.
- 4) We have added more information to figure legends, as appropriate, to enhance reader's understanding.

Reviewer #2 (Remarks to the Author)

In this paper, authors report that phosphorylated phosphoglycerate kinase 1 (PGK1) is involved in EGFR transport to lysosomes for degradation, that PGK1 binds a diLeu motif that had been previously identified in EGFR cytoplasmic domain and that PGK1 S203 phosphorylation stimulates EGFR degradation without affecting EGFR-PGK1 association. They also report that the recruitment of PGK1 to endosomes depends on PIP2 and Hrs. Authors conclude that their study reveals the existence of an unexpected coat function for a metabolic enzyme, and improves our knowledge of EGFR transport to lysosomes. The topic of the paper is clearly interesting since little is known about the direct role of metabolic enzymes in trafficking, in particular in the downregulation of EGFR. However, some serious shortcomings dampen my enthusiasm for this study.

Main comments

- 1) The observations that PGK1 binds EGFR and that this binding may depend on the presence of an intact diLeu motif in EGFR tail does not demonstrate that PGK1 is a coat adaptor. This is a serious overstatement. What is the coat, and which sorting step does it mediate?

Coat complexes couple two major functions in mediating intracellular transport, bending membrane to generate transport carriers and binding to cargoes for sorting into these carriers. The ESCRT complex acts as a coat complex for endosomal transport to the lysosome, as some of its components bind to cargoes and others bend membrane. We have found that PGK1 binds to EGFR to promote its transport to the lysosome. We also find that this role of PGK1 depends on its binding to Hrs, which is a component of the ESCRT complex. Collectively, these observations point to PGK1 acting as a coat adaptor by being an additional component of the ESCRT complex, acting to bind specific cargoes such as the EGFR.

With respect to the specific step(s) of EGFR transport that PGK1 acts, our results suggest two stages. One stage is suggested by our EM finding that si-PGK1 reduces the total level of EGFR at the late endosome (left graph, ED Fig 6e). Thus, when coupled with our other findings that PGK1 has significant association with EGFR after 15 minutes of EGF stimulation (ED Fig 4b), and EGFR localization peaks at the early endosome at this time point (ED Fig 5a), PGK1 is likely to act at the early endosome to promote EGFR transport to the late endosome.

A second stage is suggested by a new result that we have generated in addressing a comment below by this reviewer (see comment #5). We have found by quantitative EM that si-PGK1 affects the ratio of EGFR levels between the internal vesicles and the outer membrane of the late endosome, with si-PGK1 reducing this ratio (new right graph, ED Fig 6e). This finding suggests that PGK1 also promotes the sorting of EGFR from the outer membrane to the internal vesicles of the late endosome.

We thank the reviewer for asking us to clarify the role of PGK1 in EGFR transport and have added these clarifications to the revised manuscript (see the Discussion section), which has further enhanced the presentation of our work.

Does PGK1 binding to EGFR (diLeu motif) and to Hrs (regulated by PGK1 S203 phosphorylation, Fig 6c-d) correspond to the same sorting step? Or to sequential steps? Authors argue that PGK1 S203 phosphorylation targets PGK1 primarily to late endosomes (ED Fig 4). Does it mean that PGK1 binds EGFR in late endosomes? What is the relationship between these mechanisms?

We can detect PGK1 interacting with EGFR at 15 minutes after EGF stimulation (ED Fig 4b). At this time point, EGFR has its peak localization at the early endosome (ED Fig 4c). Thus, these observations suggest that the interaction between PGK1 and EGFR starts at the early endosome. We also detect a significant interaction between PGK1 and Hrs at 15 minutes after EGF stimulation (Fig 6a), suggesting that this interaction also starts at the early endosome. However, a further observation predicts that PGK1 interacts first with Hrs and then with EGFR at the early endosome, which is that siRNA against Hrs prevents PGK1 from interacting with EGFR (Fig 5d). Such stepwise interaction would also be consistent with coat adaptors needing to be recruited from the cytosol to membranes before they can interact with cargoes.

While PGK1 starts to interact with EGFR at the early endosome, we have further elucidated that their peak interaction occurs at 30 minutes after EGF stimulation (ED Fig 4b), which corresponds to when EGFR has peak localization at the late endosome (ED Fig 4d). The significance of this enhanced interaction between PGK1 and EGFR is suggested by the further observation that PGK1 is important for the sorting of EGFR into the internal vesicles of the late endosome (new right graph, ED Fig 6e).

We note that these interpretations are also consistent with our discussion above on how PGK1 promotes EGFR transport.

2) Information in text and legends is often missing or so limited that it is not easy to understand how certain experiments were performed. This is not acceptable. Some examples:

- it is often not clear whether cells were treated with EGF, and if so at what concentration and for how long (eg Fig 3a and 6). Is EGFR endocytosis (Fig 1g-h) always elicited after treatment with 100 ng/ml EGF for 1h at 4°?

We have now revised the figure legends to provide the requested details. They are highlighted in blue in the figure legends. We have also revised the Methods section (under “Cell culture and EGF treatment”) to clarify that stimulation involves EGF added at 100 ng/ml, and when the time course is not provided in the figure, the duration of EGF treatment is 1 hour at 37°C.

- The Legend of each panel of Fig 1a-j says: “Statistics is shown for a representative experiment examining the x-min time-point”? Does it mean that error bars are shown for n=1? The same comment applies elsewhere.

For all transport assays, such as those in Fig 1, we show a representative experiment from multiple independent experiments, with the total number of independent experiments indicated by “n= ”. For each experiment, we examined 10 cells for each time point, and statistics was performed for each of these time points. We have now revised the Methods section (under “Cell-based transport assays”) to add this clarification.

- How were PLA experiments done? For example, in ED Fig 2, were cells treated with EGF? What are the PLA puncta? Were cells labeled with antibodies to EGFR or to myc tag? Similarly, it is not possible to evaluate the PLA analysis in Fig 4f, 5e and 6d.

We wish to clarify that all PLA experiments were performed with EGF stimulation, as EGF stimulation is needed for PGK1 recruitment to endosomal membranes for association with EGFR. In the previous ED Fig 2g (which is now ED Fig 3d in the revised manuscript), the puncta tracks the association of PGK1 with EGFR by using primary antibodies directed against endogenous PGK1 and to the myc epitope of the transfected EGFR-myc. We have now revised the legend of this figure to clarify this point. We have also revised the figure legends of Fig 4f, 5e, and 6d to specify what protein associations are being tracked by the PLA puncta. These changes are again highlighted in blue.

- What tag is used for the wt and mutant form of PGK1 in Fig ED 2b? Also, the legends and text provide no information on how PGK1 is labeled in ED Fig 4. Do the panels show endogenous PGK1 labeled with antibodies? Or tagged, overexpressed PGK1? How are EEA1, RAB7 etc.. labeled? Are these confocal images?

We used myc-tagged forms of PGK1 for rescue experiments shown in previous ED Fig 2b, now ED Fig 2d. We have now added this clarification to the figure legend. In the case of previous ED Fig 4, now ED Fig 5 a-c and g/h, colocalization analysis was done using antibodies against endogenous PGK1, EEA1, Rab7 and Lamp1. We have also revised the legend of these figures to include these clarifications.

- In the reconstitution experiments in Fig 7i, cellular membranes prepared by subcellular fractionation were incubated with liposomes containing PIP2 or PI(3,4)P2 to modulate the phosphoinositide content of the membranes. How was PIP2 or PI(3,4)P2 delivered to membranes? By liposome fusion? If so, how was this done? How were liposomes then separated from membranes so that PGK1 recruitment could be studied?

We thank the reviewer for this comment, as it has led us to realize that we had provided an outdated description for how specific lipids were delivered to endosomal membranes. In pilot studies, we had delivered specific lipids to endosomal membranes through liposome fusion. However, precisely due to the issue raised by the comment, we subsequently took a different approach. Specific lipids can be delivered into cells simply by incubating them in the presence of albumin-containing medium (PMIDs: 25104391 and 31363100). We took this approach and then isolated endosomal membranes from the incubated cells for the reconstitution

experiments. We have now provided the description for this updated approach in the revised Methods section (under “Reconstitution of PGK1 recruitment to endosomal membrane”). Moreover, we have performed a new experiment to confirm that this approach results in lipids being delivered to endosomal membranes. This involves generating liposomes that contain NBD-labeled PC along with the specific lipid of interest, and then feeding these liposomes to cells. After isolating endosomal membranes from these cells, we detected NBD labeling of isolated membranes (new ED Fig 8i).

3) Is PGK1 really present on endosomes / lysosomes containing EGFR? Unfortunately, the analysis of PGK1 subcellular distribution by immunofluorescence is not convincing. In ED Fig 4 and all other figure, PGK1 shows a reticular, grainy pattern across the cytoplasm (excluding the nucleus), reminiscent of cytosol labeling or of background (does PGK1 staining disappear after PGK1 KD?). Moreover, this PGK1 labeling pattern is clearly different from the punctate patterns of EEA1, RAB7, LAMP1 or Hrs. How can PGK1 show 50-80% colocalization with these markers?? This is simply not possible, and colocalization of PGK1 with any of these markers seems fortuitous. The same comments apply to ED Fig 5 and ED Fig 6a, b, c and Fig 6b and 6e.

As PGK1 is a cytosolic protein, and only a fraction of PGK1 participates in EGFR transport, staining for PGK1 is predicted to have both an endosomal pattern and a cytosolic pattern. Nevertheless, we have now further documented the specificity of PGK1 staining through three lines of evidence. First, we find similar staining pattern using antibodies against PGK1 obtained from different vendors (new ED Fig 5e). Second, we find that GFP-tagged PGK1 also exhibits similar pattern as staining by antibodies (new ED Fig 5f). Third, we find that PGK1 staining is abolished by siRNA against PGK1 (new ED Fig 5d).

We also wish to point out that our confocal results show 30-55%, rather than 50-80%, PGK1 colocalization with endosomal markers (shown in previous ED Fig 4a-c, 5a-c, 6a-c, and now ED Fig 5a-c, 6a-c, and 7b-d). This degree of colocalization can be explained by the coat function of PGK1 requiring a higher protein level than its glycolytic function, as the former is predicted to require stoichiometric amount while the latter would only need catalytic amount.

4) The experiments that monitor TfR degradation in the presence of ferric ammonium citrate are not convincing (Fig 1j), given the very long time-course of these experiments (24h). Since protein synthesis goes on, these experiments monitor the total cellular levels of TfR over 24h with/without ferric ammonium citrate, and not TfR transport to the lysosomes for degradation.

We have now performed another experiment that tracks specifically the endocytic pool of TfR. This involves labeling the surface pool of TfR with fluorescence-labeled Tf and then tracking the fate of endocytic TfR by its colocalization with a lysosome marker (Lamp1). The result shows that surface TfR is transported to the lysosome upon FAC treatment, which is not affected by si-PGK1 (new ED Fig 1j).

5) The immuno-EM data in ED Fig 5d are not convincing. It is not possible to distinguish intraluminal vesicles in these micrographs. Neither is it clear what are the gold particles. Finally, the quantification shows that the total number of anti-EGFR gold particles is reduced after PGK1 KD and not the distribution of gold on limiting vs. intraluminal membranes. This suggests that PGK1 is involved in EGFR transport towards MVBs and not

EGFR incorporation into internal vesicles. Finally, the y-axis of panel 5D should not be EGFR per MVB but gold particles per profile.

We wish to clarify that the EM images in previous ED Fig 5d, now ED Fig 6e, show late endosomes, rather than entire cells, and the internal vesicles can be identified by the open space of their lumen. We have now also added arrows to highlight the gold particles in these images (see revised ED Fig 6e).

Moreover, as requested, we have quantified the level of gold particles on the internal membranes versus that on the outer membrane of the MVBs, and find that si-PGK1 reduces this ratio (new right graph, ED Fig 6e). This finding suggests that PGK1 promotes the sorting of EGFR from the outer membrane of the late endosome to the internal vesicles of this compartment. We thank the reviewer for requesting this additional analysis, as it has provided further insight into how PGK1 acts in EGFR transport.

Furthermore, as requested, we have re-labeled the y-axis of the graphs to indicate gold particles per profile.

Other comments

1) Authors mention that cargo proteins are sorted into internal vesicles of the late endosome (Multivesicular Body, MVB) for targeting to lysosome or remain at its delimiting membrane for recycling to the cell surface. This is inaccurate: i) most recycling receptors are sorted in early endosomes in order to recycle to the cell surface (e.g. TfR used in the paper); ii) ESCRT-mediated sorting begins in early endosomes, and ESCRT-I to III are primarily found on early endosomes, including Hrs, which is used in the paper (see for example Wenzel Stenmark Raiborg Nat Comm 2018). This point is important for the discussion.

We have now corrected that recycling proteins are sorted at the early endosome for return to the cell surface (see revised Introduction section). We have also added that the ESCRT complex initiates their action at the early endosome in coupling membrane bending and cargo sorting, which ultimately results in the generation of internal vesicles at the late endosome and the sorting of lysosome-bound cargoes into these vesicles (see revised Introduction and Discussion sections).

2) In Fig 1g, siPGK1 delays EGFR transport to LAMP1 by 20min. This is not a very strong inhibition. Also, how much is PGK1 overexpressed in Fig 1h and Fig 2d-e?

We wish to point out that the delayed EGFR transport induced by si-PGK1 affects EGFR-induced signaling (see Fig 2c). Thus, this delay has physiologic significance. With respect to PGK1 overexpression, the total PGK1 level is shown in Fig 2d, in which the transfected PGK1-myc, which migrates at a higher MW than the endogenous PGK1, is expressed at similar level as the endogenous PGK1. This observation suggests that the total PGK1 level is roughly doubled by the overexpression.

3) PGK catalyzes the first ATP-generating step in glycolysis. Are ATP levels altered after PGK1 depletion with siRNAs or PGK1 overexpression?

As requested, we have examined the cellular ATP level and find that altering PGK1 level has a modest effect on this level (new ED Fig 3f). An explanation for why siRNA against PGK1 has a more modest effect on ATP level than on EGFR transport is suggested by the consideration that PGK1 acts catalytically in glycolysis, and thus only a modest level is needed to support glycolysis. In contrast, reducing PGK1 level would have a more dramatic effect on EGFR transport, because cargo binding requires stoichiometric level of coat adaptors.

4) Authors argue that EGFR degradation is reduced by anti-PGK1 siRNAs (Fig 2a). If so, I do not understand why the total cellular amounts of EGFR are similar in control cells and in KD cells (Fig 2a, time 0).

We wish to point out that, when normalized to the level of tubulin, EGFR level is elevated in the KD cells. We have now provided this quantitation as a figure (new ED Fig 2c).

5) Fig 2a shows a representative experiment of EGFR degradation in time with 2 siRNAs and n=3. Which one of the two siRNAs is used for quantification in Fig 2b-c, since each graph shows 3 data points?

In Fig. 2b-2c, we quantified the effects of using siRNA #1 for the 120-minute time point. We have now added this clarification to the legend of these figures.

6) Authors report that wt or catalytically dead (T378P) PGK1 can be expressed in cells after PGK1 KD, in order to obtain physiological levels of either protein (Fig 2b). However, as I understand it, EGFR degradation is not measured after PGK1 KD followed by re-expression, but after overexpression of wt or catalytically dead PGK1 (Fig 2h and ED Fig 2c-d). Is this correct? If so, authors cannot conclude that the catalytic activity of PGK1 is not required for EGFR transport. Please clarify.

We wish to clarify that, in Fig 2h and previous ED Fig 2c/d (now ED Fig 2e/f), we knocked down PGK1 followed by rescue with myc-tagged PGK1. Thus, these experiments do not involve overexpression. We have also previously confirmed that this rescue achieves physiological levels of PGK1, as the transfected PGK1-myc is expressed at levels similar to that of endogenous PGK1 (previous ED Fig 2b, now ED Fig 2d).

7) It is not clear how the quantitative analysis of EGFR distribution was carried out (ED Fig 2a, ED Fig 3c-e, Fig 2h, 3d, 4g). Did authors quantify the number of structures that contain a given markers or the intensity of the marker? This is important since treatments that have been used (e.g. RNAi) may change the number of labeled structures (e.g. endosome fragmentation / coalescence).

For the quantitative colocalization analyses in these figures, we used the Mander's colocalization method, which is insensitive to changes in the intensity of the endosomal markers. We have also confirmed that si-PGK1 does not affect the number of labeled organelles, as exemplified by Lamp1 staining that tracks lysosomes (new ED Fig 2b).

8) The effect of the MEK inhibitor UO126 is not very convincing in Fig 4a. Also, ED Fig 4a-c indicates that the sum of the percentages of EGFR or PGK1 that colocalize with endosomal markers is greater than 100%.

Authors conclude that this is due to the fact that endosomal markers exhibit overlapping distributions (ED Fig 4d). That may be, but I find it very surprising that 40% of RAB7 colocalizes with EEA1 (ED Fig 4D). A simpler interpretation is that the labeling of the two markers is clustered in the same region, and thus overlaps in the same confocal volume (even if present on different membranes).

With respect to Fig 4a, we have now quantified the result and confirm that U0126 inhibits PGK1 recruitment appreciably (see new graph added to Fig 4a).

With respect to the previous ED Fig 4a-c (now ED Fig 5a-c), we note that a previous EM study has demonstrated that markers of the early and late endosomes have overlapping distributions (PMID: 34817533). Nevertheless, we also appreciate the explanation provided by the reviewer, as light-based microscopy cannot distinguish markers on two different membranes when they are within 200 nm. As both explanations can contribute to explaining why EGFR and PGK1 show greater than 100% total colocalization to endosomal markers, we have now included both explanations in the revised manuscript (see text added to the description of ED Fig 5g/h).

9) The data showing that EGF stimulation promotes the interaction between PGK1 and Hrs (Fig 6a) are not really convincing. The effects are small and, without quantification, it is not possible to determine whether these effects are significant.

As requested, we have performed quantitation on the interaction between PGK1 and Hrs, and find that EGF stimulation enhances this interaction appreciably (see new graph added to Fig 6a).

10) Authors state that endosomal membranes are analyzed in Fig 6F (and elsewhere in the paper). From the Methods, it is not clear whether these membranes were obtained after centrifugation at 100'000xg for 1h or after sedimentation onto a step gradient. In any case, these fractions are not endosomal membranes (Fig 6f), but relatively crude membrane fractions (e.g. do these fractions contain both early and late endosome markers?). Also, an independent equal loading control should be shown (not EGFR).

We wish to clarify that the endosomal membrane fraction was obtained after centrifugation using a sucrose gradient. Moreover, the isolated fraction is meant to be enriched in endosomal membranes rather than being completely pure, as enriched fractions are sufficient for the purpose at hand. Furthermore, we have assessed the enriched fraction for different organelle markers and find that it contains EEA1 (early endosome marker) and CD63 (late endosome marker), but not calnexin (ER marker) or COX4 (mitochondria marker) (new ED Fig 6d). We have also performed further immunoblotting of the enriched fraction to detect Lamp2 in showing similar loading across different conditions in Fig 6f (see revised Fig 6f).

11) The quantification in Fig7c shows that approx. 70% EGFR colocalizes with LAMP1 under control conditions, but a lot of red structures (EGFR) do not colocalize with green (LAMP1).

Different confocal slices can vary in colocalization. For this reason, quantitation provides a more accurate assessment of colocalization. We have also provided a different set of representative primary images to show better colocalization between EGF and Lamp1 (see revised Fig 7c).

12) What is the subcellular distribution of PIP5K1A with or without stimulation with EGF? Is PIP5K1A found on endosomes containing EGFR?

As requested, we have assessed the colocalization of PIP5K1A with endosomal EGFR (tracked by fluorescence-labeled EGF added initially to the cell surface) in a time course experiment, and detect colocalization across the endosomal compartments trafficked by endocytic EGFR (new ED Fig 7a).

Reviewer #3 (Remarks to the Author):

The epidermal growth factor receptor (EGFR) is responsible for signalling events at the plasma membrane, in endosomes and in the nucleus that lead to modulation of vital cellular processes such as proliferation and differentiation. Therefore understanding the intracellular sorting mechanism leading to its downregulation is of crucial importance. Here, Chu et al report a novel endosomal adaptor, PGK1 that is recruited to endosomal membranes upon its phosphorylation and recognizes a dileucine motif on EGFR for targeting to the lysosomal system. The authors find that this process is mediated by PGK1 interaction with Hrs and PI(4,5)P2. The work is overall significant to the field and provides substantial evidence that supports the claims. I hope that the comments below will help improve the manuscript:

1. PGK1 is a cytosolic protein, therefore colocalization studies are difficult to interpret. The staining of PGK1 in Fig.6e and ED Fig.4 and ED Fig.5 reveals a homogeneous distribution of the protein in the cytosol and even in the nucleus in some instances. The images do not convincingly show the recruitment or an enrichment of PGK1 in endosomal structures. Perhaps this could be better inferred from the gray images of the individual channels or by employing a different assay altogether. Positive/negative controls are missing.

We agree that it may be difficult to detect the membrane distribution of a cytosolic protein when only a fraction is recruited to membranes. However, we do see colocalizations in the primary images of these figures, which are better visualized in the insets of these figures. Moreover, precisely because of the issue mentioned, colocalization analysis using confocal software should be superior to visualization by eye. Thus, we have also provided quantitative colocalization results in these figures.

We have also performed additional studies to confirm the specificity of PGK1 staining. First, we find similar PGK1 staining patterns using antibodies obtained from different vendors (new ED Fig 5e). Second, taking an approach independent of antibodies, we find that GFP-tagged PGK1 exhibits similar pattern of distribution (new ED Fig 5f). Third, we find that antibody staining of PGK1 can be significantly reduced by siRNA against PGK1 (new ED Fig 5d). We further note that PGK1 has been found to have nuclear function (PMID 30392930). Thus, some PGK1 staining in the nucleus would be expected.

With respect to a different assay to confirm PGK1 recruitment to endosomal membranes, we note that endocytic EGFR marks the endosomal compartments, and we have found that PGK1 associates with EGFR upon EGF stimulation (Fig 3j and ED Fig 4b) and upon PGK1 phosphorylation (Fig 4e/f).

2. The high percentage of colocalization between PGK1 S203D mutant and Hrs in Figure 6.e is misleading as is it clear from the microscopy images that most of the protein is still in the cytosol and not in endosomes.

We apologize for having provided over-exposed images, which leads to the impression that the proteins are mostly cytosolic. We have now provided images with more appropriate exposure, and the proteins show more punctate distributions consistent with endosomal localization (revised Fig 6e).

3. Significance test were only performed for specific time points throughout the manuscript and indicators should mark only these, otherwise is misleading.

We have now indicated in the figure legend the time points that statistics are done.

4. The specific time points for which quantification was performed should be mentioned in the Figures throughout the manuscript (e.g. Fig.1j, Fig.2 b,c,e,f etc).

We have now revised the legend of figures that have time-course analysis to indicate the time points that quantitation and statistics were done.

REVIEWER COMMENTS

Reviewer #1 (Remarks to the Author):

The authors have performed a series additional experiments addressing all my major concerns. The manuscript is suitable for publication.

Reviewer #2 (Remarks to the Author):

In this revised version, the authors have done considerable work to respond to the reviewers' comments and have added a significant amount of new data. However, some important points are still unclear and, unfortunately, raise questions that cast doubt on the significance of the observations reported in the paper.

1) Authors mention that PGK1 interacts first with Hrs and then with EGFR at the early endosome (Hrs KD prevents PGK1-EGFR interactions, Fig 5d), and argue that this stepwise interaction is consistent with coat adaptors being recruited from the cytosol to membranes before they can interact with cargoes. However, this experiment says nothing about the time course of the reaction: it only shows that PGK1 does not interact with EGFR without Hrs.

2) In the rebuttal letter, authors mention that they had provided an outdated description for how specific lipids were delivered to endosomal membranes (Fig 7i), and that lipids were in fact delivered into cells after incubation with BSA-containing medium. I find this rather surprising because lipid delivery via liposome fusion with purified endosomes in vitro (old version) or via lipid addition to living cells via BSA (new version) are very different techniques. In addition, the description of the experiment is still different in the Methods section, where it is stated that cells were treated with liposomes containing PIP2 or PI3,4P2 (generated in the presence of BSA). This all is very confusing.

More important, the PI3,4P2 and PIP2 content of the fractions should be measured, to show that PI3,4P2 or PIP2 are present in these membranes. Also, Fig 7i is not really convincing. There is a bit more PGK1 in the PIP2 lane (50 nM), but there is also a bit more EGFR. Finally, authors have added a new experiment (ED Fig 8i) to show that lipids are delivered to endosomal membranes. Cells were treated with liposomes labeled with NBD-PC, and, after homogenization and fractionation, NBD was detected in isolated membranes. However, this experiment does not say anything about the presence of PI3,4P2 and PIP2 in

endosomes. For example, what is the evidence that PI3,4P2 and PIP2 are not hydrolyzed during homogenization and preparation of the membranes?

3) It is very nice that the authors could convincingly demonstrate that the antibody used is specific (ED Fig 5d-f). However, I am still not convinced that the colocalization in Fig ED Fig 6 is meaningful. The green pattern of PGK1 and the red pattern of each marker (EEA1, Rab7 etc..) are clearly different. PGK1 is cytosolic: this is very nicely illustrated in new ED Fig 5d-f, where organelles appear as black shadows in the white sea of the cytosol. The labeling of the organelle's rim, if any, is very faint in all panels of ED Fig 5. PGK1 colocalization with any organelle marker in the 40-60% range (see panels) is simply not possible! Only a tiny fraction of PGK1 may be found around the rim of each organelle. Also, the legend does not correspond to the Fig. It is stated that the Fig compares 15-min versus 30-min time points, but the Fig shows 3 time-points. In the end, is PGK1 found on endosomes?

Minor point

1) I am still not convinced that the term coat adaptor is appropriate, particularly in the title. Several factors have been proposed to play a role in ESCRT membrane association. For example, the term adaptor has been used by McCullough et al. (Annu Rev Cell Dev Biol, 2019) for proteins that recruit early-acting ESCRTs, like CEP55 at the midbody or ESCRT-0 (HRS/STAM) on endosomes. Is HRS/STAM an ESCRT adaptor? The precise role of PGK1 (or ESCRT-0) is not sufficiently clear. I also wish to add that KD of PGK1 only delays EGFR transport to LAMP1 by 20min (Fig 2g). This is a rather modest effect for an adaptor involved in sorting to lysosomes.

2) The data in ED Fig 6E are potentially interesting. However, I find it difficult to identify gold particles in the micrograph and to discriminate them from the black stuff that is all over, presumably a precipitate.

RESPONSES (manuscript # NCOMMS-22-46410B)

In response to our revision, one reviewer has provided further comments. They are listed below along with our point-by-point responses, which include additional experiments that we have performed. Some responses also involve revising the manuscript text. These changes are highlighted in blue to facilitate their detection.

Reviewer #1 (Remarks to the Author):

The authors have performed a series additional experiments addressing all my major concerns. The manuscript is suitable for publication.

Reviewer #2 (Remarks to the Author):

In this revised version, the authors have done considerable work to respond to the reviewers' comments and have added a significant amount of new data. However, some important points are still unclear and, unfortunately, raise questions that cast doubt on the significance of the observations reported in the paper.

We thank the reviewer for acknowledging that we have done considerable work in responding to previous reviewers' comments. Below, we respond to further comments that have been raised in light of our revision.

1) Authors mention that PGK1 interacts first with Hrs and then with EGFR at the early endosome (Hrs KD prevents PGK1-EGFR interactions, Fig 5d), and argue that this stepwise interaction is consistent with coat adaptors being recruited from the cytosol to membranes before they can interact with cargoes. However, this experiment says nothing about the time course of the reaction: it only shows that PGK1 does not interact with EGFR without Hrs.

- As we had found that EGF stimulation recruits PGK1 to endosome membrane for interaction with EGFR and Hrs, we had previously considered two temporal possibilities for how PGK1 recruitment occurs: i) PGK1 first interacts with EGFR and then with Hrs, or ii) PGK1 first interacts with Hrs and then with EGFR. Thus, because targeting against Hrs prevents PGK1 from interacting with EGFR, we concluded that the second scenario was more likely. However, the comment points out that our data does not rule out a third possibility, which is that PGK1 interacts simultaneously with EGFR and Hrs. We thank the reviewer for pointing this out and have revised the Discussion section to mention this additional possibility.

2) *In the rebuttal letter, authors mention that they had provided an outdated description for how specific lipids were delivered to endosomal membranes (Fig 7i), and that lipids were in fact delivered into cells after incubation with BSA-containing medium. I find this rather surprising because lipid delivery via liposome fusion with purified endosomes in vitro (old version) or via lipid addition to living cells via BSA (new version) are very different techniques. In addition, the description of the experiment is still different in the Methods section, where it is stated that cells were treated with liposomes containing PIP2 or PI3,4P2 (generated in the presence of BSA). This all is very confusing.*

- We apologize for the confusion. As noted by the comment, lipids can be delivered into cells either by feeding in the presence of BSA or by incorporating into liposomes followed by incubation with cells. Through serendipity, we had discovered that by incorporating PI(4,5)P2 into liposomes in the presence of BSA followed by incubation with cells, this approach led to even more efficient lipid delivery to endosomal membranes. Thus, this approach was described in the Methods section.

More important, the PI3,4P2 and PIP2 content of the fractions should be measured, to show that PI3,4P2 or PIP2 are present in these membranes. Also, Fig 7i is not really convincing. There is a bit more PGK1 in the PIP2 lane (50 nM), but there is also a bit more EGFR. Finally, authors have added a new experiment (ED Fig 8i) to show that lipids are delivered to endosomal membranes. Cells were treated with liposomes labeled with NBD-PC, and, after homogenization and fractionation, NBD was detected in isolated membranes. However, this experiment does not say anything about the presence of PI3,4P2 and PIP2 in endosomes. For example, what is the evidence that PI3,4P2 and PIP2 are not hydrolyzed during homogenization and preparation of the membranes?

- We have now pursued an ELISA-based approach to detect PI(4,5)P2 and PI(3,4)P2. We first confirmed that siRNA against PIP5K1A reduced the level of PI(4,5)P2 but not that of PI(3,4)P2 on endosomal membrane isolated from the siRNA-treated cells. See new ED Fig 8i. We then found that PI(4,5)P2 level on PIP5K1A-depleted endosomal membrane was increased when the siRNA-treated cells were fed with PI(4,5)P2 but not with PI(3,4)P2. Conversely, the level of PI(3,4)P2 but not that of PI(4,5)P2 was increased on endosomal membrane when cells were fed with PI(3,4)P2. See new ED Fig 8j.

- We have also quantified the data in Fig 7i to confirm that PGK1 recruitment to PIP5K1A-depleted endosomal membrane was significantly enhanced at higher levels of PI(4,5)P2 used for functional rescue.

3) *It is very nice that the authors could convincingly demonstrate that the antibody used is specific (ED Fig 5d-f). However, I am still not convinced that the colocalization in Fig ED Fig 6 is meaningful. The green pattern of PGK1 and the red pattern of each marker (EEA1, Rab7 etc..) are clearly*

different. PGK1 is cytosolic: this is very nicely illustrated in new ED Fig 5d-f, where organelles appear as black shadows in the white sea of the cytosol. The labeling of the organelle's rim, if any, is very faint in all panels of ED Fig 5. PGK1 colocalization with any organelle marker in the 40-60% range (see panels) is simply not possible! Only a tiny fraction of PGK1 may be found around the rim of each organelle. Also, the legend does not correspond to the Fig. It is stated that the Fig compares 15-min versus 30-min time points, but the Fig shows 3 time-points. In the end, is PGK1 found on endosomes?

- With respect to the comment that ED Fig 6 does not show significant colocalization of PGK1 with endosomal markers, we respond with two explanations that are complementary. First, representative images are single confocal slices, which do not capture the total degree of colocalization. Second, judging colocalization by eye can underestimate the degree of colocalization. With respect to the latter, the eye best appreciates colocalization as manifested by yellow. However, lesser degrees of colocalization that are manifested by shades of orange are less appreciated by the eye but would not escape software detection. Importantly, we further note that, besides confocal microscopy, we had examined endosomal membranes by immunoblotting and found that nearly 50% of PGK1 is recruited to endosomal membranes. See Figs 4a and 4b. Thus, there is more than just a tiny fraction of PGK1 recruited onto endosomal compartments.

- With respect to the comment that PGK1 colocalization with any particular endosomal marker in the 40-60% range is not possible, we agree, and have explained previously that the endosomal markers show overlap, which we had documented (previous ED Figs 5g and 5h, now ED Figs 4l and 4m). Thus, the true degrees of PGK1 colocalizing with these markers are expected to be less than that shown. Nevertheless, we appreciate the comment that has pointed out that our current way of expressing the colocalization values is misleading. Thus, we have now expressed the colocalization values as Pearson coefficients, which simply assesses the degree of colocalization rather than also calculating for the fraction of PGK1 colocalizing with endosomal markers. Notably, this serves the purpose of these experiments, as they are intended to determine the time point when EGFR shows the greatest distribution at a particular endosomal compartment. See revised ED Figs 4g/4i/4k, 5a/5b/5c, and 7b/7c/7d. We also took this approach in assessing the colocalization of PGK1 with Hrs. See revised 6a/6b/6c.

- We also wish to clarify that our comparison of two time points refers to statistical analysis, not the quantitation data on colocalization which are provided for three time points.

Minor point

1) *I am still not convinced that the term coat adaptor is appropriate, particularly in the title. Several factors have been proposed to play a role in ESCRT membrane association. For example, the term adaptor has been used by McCullough et al. (Annu Rev Cell Dev Biol, 2019) for proteins that recruit early-acting ESCRTs, like CEP55 at the midbody or ESCRT-0 (HRS/STAM) on endosomes.*

Is HRS/STAM an ESCRT adaptor? The precise role of PGK1 (or ESCRT-0) is not sufficiently clear. I also wish to add that KD of PGK1 only delays EGFR transport to LAMP1 by 20min (Fig 2g). This is a rather modest effect for an adaptor involved in sorting to lysosomes.

- We appreciate the reviewer's point, as the term adaptor can have different meaning depending on the context. We had meant that PGK1 acts as an adaptor that links EGFR to Hrs (a component of the ESCRT complex) in explaining how the endosomal transport of EGFR is enhanced. In other transport pathways, factors that play such a role have also been referred as cargo adaptors. Thus, we have now revised the manuscript to refer to PGK1 as a cargo adaptor.

- We also wish to point out that siRNA treatment reduces, rather than completely abolishes, a protein level, which explains the quantitative effect of siRNA against PGK1 on EGFR transport. Importantly however, this quantitative reduction has physiologic significance, as EGFR signaling is affected, which is reflected by Erk activation. See Fig 2c.

2) The data in ED Fig 6E are potentially interesting. However, I find it difficult to identify gold particles in the micrograph and to discriminate them from the black stuff that is all over, presumably a precipitate.

- We have now re-performed this experiment using larger gold particles (15 nm gold rather than 10 nm gold), which enables better distinction of gold particles from the uranyl staining of proteins. See revised Fig 6e.

REVIEWERS' COMMENTS

Reviewer #2 (Remarks to the Author):

The authors have addressed my comments in a satisfactory manner.